# Recurrent Structural Policy Gradient
# for Partially Observable Mean Field Games

**Clarisse Wibault** [1] [2]  **Johannes Forkel** [1]  **Sebastian Towers** [1]  **Tiphaine Wibault** [3]  **Juan Duque** [4]  **George Whittle** [2]
**Andreas Schaab** [5]  **Yucheng Yang** [6]  **Chiyuan Wang** [7]  **Maike Osborne** [2]  **Benjamin Moll** [† 8]  **Jakob Foerster** [† 1]

## Abstract

Mean Field Games (MFGs) provide a principled framework for modelling interactions in large population systems. However, algorithmic progress has been limited since model-free methods are high variance and exact methods scale poorly. Recent Hybrid Structural Methods (HSMs) reduce variance while maintaining tractability by leveraging low-dimensional individual state and action spaces and known transition dynamics to compute the exact expected return conditioned on Monte Carlo rollouts of common noise. However, HSMs have not been extended to partially observable settings. We propose *Recurrent Structural Policy Gradient* (RSPG), the first history-aware HSM for MFGs with public partial information. RSPG achieves an order-of-magnitude faster convergence than model-free RL methods while learning history-aware behaviour, unlike current HSMs. To facilitate research into MFGs, we also introduce MFAX, our JAX-based framework for MFGs that supports both analytic and sample-based mean-field updates. MFAX and usage examples can be found at https://clarisse-wibault.github.io/rspg/.

## 1 Introduction

Training policies in large multi-agent systems is notoriously difficult: Multi-Agent Reinforcement Learning (MARL) based methods rely on high-variance, trajectory-based sampling and therefore scale poorly as the number of agents

grows (Daskalakis et al., 2009; Matignon et al., 2012; Cui et al., 2022). But in many large population systems, such as financial markets, traffic control and communication networks, individuals only respond to the population-level distribution of other agents, rather than their individual identities (Cardaliaguet & Hadikhanloo, 2017; Yang et al., 2017b; Cabannes et al., 2021). As a motivating example, a rational investor responds to the price of an asset determined by the rest of the population more than to the isolated decisions of another trader.

Hence, assuming all agents have the same state-conditioned objective, reward, transition and observation models, analysis can be reduced to the interaction between a *stand-in agent* with policy $\pi$ and the *population distribution*, or *mean-field* $\mu_t$ over the individual state space $\mathcal{S}$. Such *Mean-Field Games* (MFGs) (Lasry & Lions, 2007; Huang et al., 2006) have found applications in epidemiology, communication systems, energy networks, finance, and macroeconomics (Elie et al., 2020a; Alasseur et al., 2020; Yang et al., 2017a; Cardaliaguet & Hadikhanloo, 2017; Moll, 2026).

In MFGs with *common noise*, uncertainty enters through aggregate shocks that affect the entire population simultaneously. While idiosyncratic noise marginalises out at the population level, common noise induces stochastic evolution of the mean-field. The resulting Markovian *aggregate state* consists of both the mean-field and the common noise $z_t$, which captures all remaining state components.

In *partially observable* MFGs, agents must infer the aggregate state $(\mu_t, z_t)$ rather than observing it directly. As summarised by Assumption 1.1, in many applications of interest, agents only receive a *shared partial observation* of the aggregate state[1]. The public partial information might be stock prices, interest rates, reported infection levels, or communication signals, for example (Cardaliaguet et al., 2019; Moll & Ryzhik, 2026; Elie et al., 2020b; Alasseur et al., 2020; Yang et al., 2017a).

The issue is that existing methods for modelling large popu-

---

[†] Equal supervision [1] FLAIR, University of Oxford [2] MLRG, University of Oxford [3] ifo Institute, LMU Munich [4] Mila, Québec AI Institute [5] UC Berkeley [6] University of Zurich [7] Peking University [8] London School of Economics. Correspondence to: Clarisse Wibault <clarisse.wibault@magd.ox.ac.uk>.

*Proceedings of the 43$^{rd}$ International Conference on Machine Learning*, Seoul, South Korea. PMLR 306, 2026. Copyright 2026 by the author(s).

[1] We remark that $o_t$ being a deterministic rather than random function of $(\mu_t, z_t)$ is without loss of generality, since for shared public observations randomness can be incorporated within $z_t$.

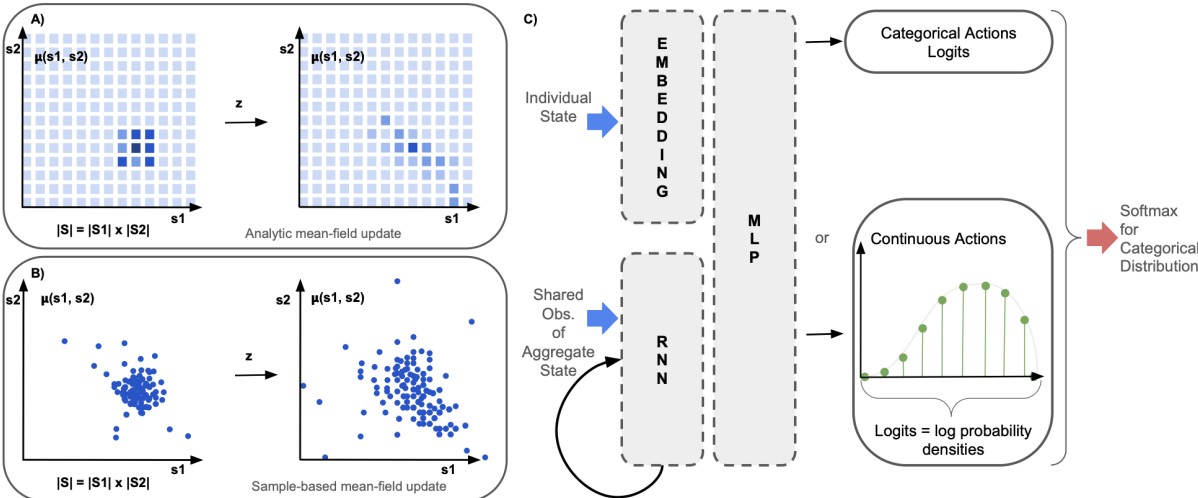

*Figure 1.* **Top left**: **analytic** mean-field updates compute exact expectations over next states. **Bottom left**: **sample-based** mean-field updates re-approximate the mean-field at each step. **Right**: RSPG architecture.

lation systems typically address only subsets of these three challenges posed by mean-field interaction, common noise, and partial observability. Deep Reinforcement Learning (RL) approaches for MFGs with common noise could trivially incorporate memory through recurrence to handle partial observability, but these methods are model-free and therefore do not exploit known system structure for variance reduction (Elie et al., 2020b). Recent Hybrid Structural Methods (HSMs) (Han et al., 2021), such as Structural Policy Gradient (SPG) (Yang et al., 2025), leverage low-dimensional individual state and action spaces (e.g. the mean-field may be defined over wealth and income in macroeconomics, infection status in epidemiology, or energy and interference levels in communication systems) to achieve lower-variance updates. But, HSMs are restricted to tabular, memoryless policies; extending them to incorporate history-dependence is challenging due to the exponentially growing dimensionality of the history space.

**Assumption 1.1.** *Structural Assumption: Public Partial Information. Agents receive a shared observation of the aggregate state, $o_t = \mathcal{U}(\mu_t, z_t)$.*

In this work, we introduce *Recurrent Structural Policy Gradient* (RSPG), a HSM that supports history-dependent policies while preserving tractability by restricting memory to the history of shared observations. Empirically, RSPG maintains the same rate of convergence as memoryless HSMs, an order of magnitude faster than model-free RL methods, while its history-awareness enables the learning of more realistic agent behaviour. We summarise our contributions as follows:

1. We provide a unified taxonomy of Dynamic Programming (DP), Reinforcement Learning (RL) and Hybrid Structural Methods (HSMs) for MFGs that is missing in the MFG literature.

2. We introduce *Partially Observable Mean Field Games with Common Noise* and highlight the associated challenges of (i) updating the mean-field; and (ii) maintaining tractability of Hybrid Structural Methods.

3. We propose *Recurrent Structural Policy Gradient* (RSPG), the first history-aware HSM. RSPG preserves the variance-reduction benefits of HSMs while enabling history-dependent behaviour.

4. We introduce MFAX, a JAX-based framework for MFGs. Unlike existing libraries, MFAX (i) distinguishes between white-box and sample-based access to transition dynamics, (ii) accelerates analytic mean-field updates through functional representations rather than matrix multiplication, and (iii) supports common noise, partial observability, and multiple initial mean-field distributions.

## 2  Related Work

The literature on MFGs spans a rich body of research. Laurière et al. (2022) survey the standard setting (excluding multiple initial distributions, common noise, and partial observability), where the mean-field evolves deterministically. Numerical methods (Achdou & Capuzzo-Dolcetta, 2010; Lauriere, 2021) can solve these problems when analytic solutions are unavailable, but are not suitable for settings with common noise. Solving the associated Master Equation (Cardaliaguet et al., 2019) has motivated the use of Deep

**Analytic Mean-Field Update in Fully-Observable Settings**

$$\mu_{t+1}(s_{t+1}) := p(s_{t+1} \mid \mu_t, z_t) = \int_{\mathcal{S}} \int_{\mathcal{A}} \mathcal{T}(s_{t+1} \mid s_t, a_t, \mu_t, z_t)\pi(a_t \mid s_t, \mu_t, z_t)\mu_t(s_t)\mathrm{d}a_t\mathrm{d}s_t, \qquad (1)$$

Learning based methods; we refer to Carmona et al. (2021); Hu & Lauriere (2023), and the references therein.

While deep RL methods have been developed for MFGs with common noise, and could be trivially extended to support partial observability, these methods are model-free (Elie et al., 2020b), treating dynamics as a black box and foregoing variance reduction from known structure. Although Dynamic Programming or model-based methods have been proposed (Perrin et al., 2020), these require enumerating realisations of common noise and associated mean-field trajectories, limiting scalability. Recent Hybrid Structural Methods (HSMs) (Han et al., 2021), such as Structural Policy Gradient (SPG) (Yang et al., 2025), leverage low-dimensional individual state and action spaces and known transition dynamics to achieve lower-variance updates, but discard available information to maintain tractability by sampling common noise. As mentioned in the introduction, the issue is that existing HSMs are limited to tabular, memoryless policies.

Most Deep Learning based algorithms for MFGs assume full observability, with policies conditioning on the local state $\pi(\cdot \mid s_t)$ (without common noise) (Perrin, 2022; Algumaei et al., 2023; Hu & Zhang, 2025; Cui & Koeppl, 2021) or both the local and aggregate state $\pi(\cdot \mid s_t, \mu_t, z_t)$ (with common noise) (Wu et al., 2025; Perrin et al., 2021). We remark that conditioning on the mean-field is only necessary in the presence of common noise. Partial observability has been considered in limited forms: Subramanian et al. (2020) restrict observations to local neighbourhoods, and Benjamin & Abate allow agents to estimate the mean-field, but both use memoryless policies. On the theoretical side, Yongacoglu et al. (2024) study independent learning in partially observable MFGs with shared observations and prove convergence for memoryless policies, while Saldi et al. (2019) consider more general observation kernels. Neither includes common noise. To our knowledge, this is the first work to address both common noise and partial observability, so we formalise the problem setting using a general observation model, and only then consider the special case of shared aggregate observations.

In Mean Field Control (MFC) (Hu & Lauriere, 2023; Carmona et al., 2021), unlike in MFGs, agents are cooperative and maximise social welfare. Note that MFC is different from the control problem within the MFG equilibrium problem, which we introduce in Section 5.

## 3 Preliminaries

**Mean Field Games (MFGs) with Common Noise.** An infinite-horizon MFG with common noise (Carmona et al., 2016; Perrin et al., 2020; Wu et al., 2025) is defined by a tuple $(p_{z_0}, p_{\mu_0}, \mathcal{Z}, \mathcal{S}, \mathcal{A}, \Xi, \mathcal{T}, R, \gamma)$, where $p_{\mu_0}$ is the distribution over initial mean-fields, $p_{z_0}$ the distribution over initial common noise, $\mathcal{Z}$, $\mathcal{S}$ and $\mathcal{A}$ respectively denote the finite common noise, individual state and action spaces, $\Xi$ the common noise transition function, $\mathcal{T}$ the individual state transition function, $R$ the reward function and $\gamma$ the discount factor. At time $t = 0$, the initial common noise $z_0$ is drawn from the distribution $p_{z_0}$ and an initial mean-field distribution $\mu_0$ over the individual state space $\mathcal{S}$ is drawn from $p_{\mu_0}$. At each timestep $t \geq 0$, agents take an action $a_t \sim \pi(\cdot \mid s_t, \mu_t, z_t)$ conditioned on their state $s_t$, the mean-field $\mu_t$ and common noise $z_t$, and transition to a new state $s_{t+1} \sim \mathcal{T}(\cdot \mid s_t, a_t, \mu_t, z_t)$. The common noise evolves according to $z_{t+1} \sim \Xi(\cdot \mid z_t)$. By conditioning on $\mu_t$ and $z_t$, the next mean-field $\mu_{t+1}$ is deterministic, as given by Equation 1. Agents receive a scalar reward $r_t = R(s_t, a_t, \mu_t, z_t)$ that is a function of their individual state, the action, and the aggregate state $(\mu_t, z_t)$.

Let $\mathrm{M}^{\pi', z_{0:t}}$ denote the mean-field sequence $\mu_{0:t} = (\mu_0, ..., \mu_t)$ that is generated from $\pi'$ and $z_{0:t}$ through Equation 1. For convenience, in the following, we absorb randomness of $\mu_0$ into the initial common noise by replacing $z_0$ with an augmented tuple $(z_0, \eta)$, where $\eta$ is an independent seed, such that $\mu_{0:\infty}$ is deterministic given $z_{0:\infty}$.

In the MFG control problem, the aim is to find a policy $\pi$ conditioned on an exogenous population behaviour induced by a fixed policy $\pi'$ that maximises

$$J(\pi, \pi') := \mathbb{E}\left[\sum_{t=0}^{\infty} \gamma^t R(s_t, a_t, \mu_t, z_t)\right], \qquad (2)$$

where $z_0 \sim p_{z_0}$, as described above, $z_0$ is augmented so $\mu_0$ is deterministic, and $s_0 \sim \mu_0$, $a_t \sim \pi(\cdot \mid s_t, \mu_t, z_t)$, $s_{t+1} \sim \mathcal{T}(\cdot \mid s_t, a_t, \mu_t, z_t)$, $z_{t+1} \sim \Xi(\cdot \mid z_t)$, $\mu_t = \mathrm{M}_t^{\pi', z_{0:t}}$. The MFG control problem is therefore equivalent to finding an optimal policy in a non-stationary single-agent MDP.

In contrast, in the MFG equilibrium problem, the mean-field sequence is endogenous and generated by the policy $\pi$ itself, such that $\mu_t = \mathrm{M}_t^{\pi, z_{0:t}}$. The aim is to find a mean-field Nash equilibrium, defined as follows.

---

**Information Box 1.**

**Dynamic Programming** based methods leverage full access to the model i.e. both the individual state dynamics $\mathcal{T}(s' \mid s, a, \mu, z)$ and common noise dynamics $\Xi(z' \mid z)$.

The induced mean-field sequence is computed analytically:

$$\boldsymbol{\mu}' = \mathbf{A}_{\mu,z}^{\pi \top} \boldsymbol{\mu}.$$

The policy is evaluated by computing the exact expected returns using the infinite-dimensional "Master-Equation":

$$V(s, \boldsymbol{\mu}, z) = \mathbb{E}_{a \sim \pi(\cdot|s,\mu,z)} \left[ R(s, a, \boldsymbol{\mu}, z) + \gamma \mathbb{E}_{s' \sim \mathcal{T}(\cdot|s,a,\mu,z), z' \sim \Xi(\cdot|z)} \left[ V(s', \boldsymbol{\mu}', z') \right] \right] \qquad (4)$$

or, in state-vectorised form:

$$\mathbf{v}_{\mu,z} = \left[ \mathbf{\Pi}_{\mu,z} \odot \mathbf{R}_{\mu,z} \right] \mathbf{1} + \gamma \mathbf{A}_{\mu,z}^{\pi} \mathbb{E}_{z' \sim \Xi(\cdot|z)} \left[ \mathbf{v}_{\mu',z'} \right], \qquad (5)$$

where $\mathbf{v}_{\mu,z}$ is an $|\mathcal{S}|$-length vector with the values $V(s, \boldsymbol{\mu}, z)$ of each individual state $s \in \mathcal{S}$, and $\mathbf{\Pi}_{\mu,z}$ and $\mathbf{R}_{\mu,z}$ are $|\mathcal{S}| \times |\mathcal{A}|$ matrices whose $(s,a)^{\text{th}}$ elements are $\pi(a \mid s, \boldsymbol{\mu}, z)$ and $R(s, a, \boldsymbol{\mu}, z)$ respectively.

Note that premultiplying by $\mathbf{A}_{\mu,z}^{\pi}$ computes the expectation over next states.

---

**Definition 3.1.** A policy $\pi^*$ and its induced population behaviour constitute a Nash equilibrium if and only if

$$\pi^* \in \text{argmax}_{\pi \in \Pi} J(\pi, \pi^*).$$

In other words,

$$J(\pi^*, \pi^*) \geq J(\pi, \pi^*) \quad \forall \quad \pi \in \Pi,$$

where $\Pi$ is the set of all policies conditioning on $(s, \mu, z)$.

Since $\mathcal{S}$ and $\mathcal{A}$ are finite, Equation 1 can be rewritten in vector-matrix form,

$$\boldsymbol{\mu}_{t+1} = \mathbf{A}_{\mu_t, z_t}^{\pi \top} \boldsymbol{\mu}_t, \qquad (3)$$

where $\mathbf{A}_{\mu_t, z_t}^{\pi}$ is a matrix whose $(s_t, s_{t+1})^{\text{th}}$ entry is $\mathbb{E}_{a_t \sim \pi(\cdot|s_t, \mu_t, z_t)} \left[ \mathcal{T}(s_{t+1} \mid s_t, a_t, \mu_t, z_t) \right]$ and $\boldsymbol{\mu}_t$ is a vector of length $|\mathcal{S}|$ such that $\min_{k=1,\ldots,|\mathcal{S}|} \boldsymbol{\mu}_t(k) \geq 0$ and $\sum_{k=1}^{|\mathcal{S}|} \boldsymbol{\mu}_t(k) = 1$.

## 4 Dynamic Programming, Reinforcement Learning & Hybrid Structural Methods

In settings with common noise, the policy is updated by computing its value and then taking a greedy step to improve that policy.

This section provides a unified taxonomy of policy evaluation using Dynamic Programming (DP), Reinforcement Learning (RL) and Hybrid Structural Methods (HSMs), respectively leveraging full, no or partial knowledge of the model.

**Dynamic Programming (DP)** based methods leverage full access to the model i.e. both the individual state dynam-

ics, and common noise dynamics (Information Box 1). By evaluating the policy using exact expected returns, the resulting Bellman equation (Equation 4) or *Master Equation* eliminates sample variance (Cardaliaguet & Hadikhanloo, 2017). However, in practice, integrating over all possible realisations of the mean-field due to different realisations of common noise renders DP intractable.

**Reinforcement Learning (RL)** based methods do not leverage access to transition dynamics (Information Box 2), instead relying on Monte Carlo samples for policy evaluation. Moreover, rather than computing the analytic mean-field update, the mean-field distribution is repeatedly re-approximated using sample-based methods (bottom left Figure 1) or by learning a function that can approximate the analytic mean-field update (Perrin et al., 2021; Inoue et al., 2021). By treating environment dynamics as a black box, RL based methods enable learning under unknown dynamics, intractably large individual state or action spaces, dense transition operators, and in partially-observable settings where the observation function depends on the individual state (Section 5). But, being fully sample-based, they suffer from high-variance value estimates, and hence high-variance policy updates.

**Hybrid Structural Methods (HSMs)** are an intermediary between DP and RL based methods as they leverage access to known individual state transition dynamics, but use Monte Carlo rollouts of the common noise (Information Box 3). By replacing sampling of next-state transitions with analytic expectations, HSMs achieve lower-variance gradient updates (and hence faster convergence) than RL based methods, but remain tractable, unlike DP.

---

**Information Box 2.**

**Reinforcement Learning** based methods do not leverage access to transition dynamics, instead sampling next states, actions and common noise.

The induced mean-field sequence is computed by repeatedly re-approximating the mean-field:

$$\boldsymbol{\mu}_t(s) \approx \frac{1}{N} \sum_{i=1}^{N} \delta_{s_{i,t}}(s). \tag{6}$$

The policy is evaluated using Monte Carlo rollouts:

$$\hat{V}(s_{i,t}, \boldsymbol{\mu}_t, z_t) \approx R(s_{i,t}, a_{i,t}, \boldsymbol{\mu}_t, z_t) + \gamma \hat{V}(s_{i,t+1}, \boldsymbol{\mu}_{t+1}, z_{t+1}) \tag{7}$$

where $a_{i,t} \sim \pi(\cdot|s_{i,t}, \boldsymbol{\mu}_t, z_t)$, $s_{i,t+1} \sim \mathcal{T}(\cdot \mid s_{i,t}, a_{i,t}, \boldsymbol{\mu}_t, z_t)$, $z_{t+1} \sim \Xi(\cdot \mid z_t)$ and the mean-field is re-approximated.

---

# 5 Partially Observable Mean Field Games with Common Noise

We formalise *Partially Observable Mean Field Games with Common Noise* (POMFGs-CN) as decision making under uncertainty, where agents receive only partial information about the aggregate state $(\mu_t, z_t)$. We remark that in POMFGs-CN, the aggregate state is not Markovian in general, but we refer to it as such for consistency.

A POMFG-CN is defined by a tuple $(p_{z_0}, p_{\mu_0}, \mathcal{Z}, \mathcal{S}, \mathcal{A}, \mathcal{O}, \Xi, \mathcal{T}, \mathcal{U}, R, \gamma)$, where $\mathcal{O}$ denotes the observation space, $\mathcal{U}$ the observation function, and $p_{z_0}, p_{\mu_0}, \mathcal{Z}, \mathcal{S}, \mathcal{A}, \Xi, \mathcal{T}, R, \gamma$ are as defined in Section 3. At each timestep $t \geq 0$, agents obtain an observation $o_t$ of the aggregate state $o_t \sim \mathcal{U}(\cdot \mid s_t, \mu_t, z_t)$. Since when $o_t$ depends on the individual state $s_t$, there is information in the Individual-Action-Observation History (IAOH) $\tau_t := (s_0, o_0, a_0, s_1, o_1, a_1, ..., s_t, o_t) \in \mathcal{H}_t := (\mathcal{S} \times \mathcal{O} \times \mathcal{A})^t \times \mathcal{S} \times \mathcal{O}$, agents take an action $a_t \sim \pi(\cdot \mid \tau_t)$ conditioned on $\tau_t$. The transition dynamics, common noise dynamics and reward function are as before.

Computing the analytic mean-field update requires keeping track of not just the current mean-field ($\mu_t$, a distribution over $\mathcal{S}$), but a distribution $\tilde{\mu}_t$ over the history space $\mathcal{H}_t$. Since $(s_0, s_1, ..., s_t) = s_{0:t}$ is a function of $\tau_t$, $\mu_{0:t} = (\mu_0, \mu_1, ..., \mu_t)$ is fully determined by $\tilde{\mu}_t$. Note that the reverse is not true, i.e. $\mu_{0:t}$ does not fully determine $\tilde{\mu}_t$. Similarly, by conditioning on $\tilde{\mu}_t$, $z_t$ and $z_{t+1}$, and writing $\tau_{t+1} = (\tau_t, a_t, s_{t+1}, o_{t+1})$, $\tilde{\mu}_{t+1}$ is deterministic, as given by Equation 12, where we use the fact that $\mu_{t+1}$ is deterministic given $(\tilde{\mu}_t, z_t)$ to find the updated mean-field for the observation kernel (Equation 11).

As in fully-observable MFGs defined in Section 3, the aim is to find a mean-field Nash Equilibrium, where the expectation is now also taken with respect to the observation kernel, and the policy conditions on the IAOH.

# 6 Recurrent Structural Policy Gradient

When policies condition on IAOHs, computing the analytic mean-field update and exact expectations required by HSMs becomes intractable. In the fully observable setting, HSMs only need to maintain a distribution $\mu_t$ over the fixed individual state space $\mathcal{S}$. Under partial observability, however, they must maintain a distribution $\tilde{\mu}_t$ over the history space $\mathcal{H}_t$. The issue is that the number of possible histories grows exponentially with time such that updating the mean-field and evaluating the policy would require enumerating an exponentially branching history tree.

Our insight is that, in many applications with public partial information (Assumption 1.1), memory can be restricted to the history of shared aggregate observations. We formalise this by characterising the relative expressiveness of the public observation channel in Assumption 6.1, which serves as a natural approximation in applications where, once an agent knows its current individual state and the public history, its past private trajectory provides negligible additional information about the aggregate state[2].

**Assumption 6.1.** *Informational Assumption: Sufficient Public Partial Information. Given $(s_t, o_{0:t})$, the IAOH $\tau_t$ contains no additional information about the aggregate state $(\mu_t, z_t)$, such that $p(\mu_t, z_t \mid \tau_t) = p(\mu_t, z_t \mid s_t, o_{0:t})$.*

In financial markets, for example, an investor's past trades provide little additional information about the state of the economy given the public history of stock prices and their current portfolio value. Similarly, an individual's history of infections reveals little about the state of an epidemic given the public history of infection rates and the individual's current health status.

---

[2]While this information equivalence does not hold universally (see the counter-example in Appendix A) it is a useful approximation in many applications of interest.

**Information Box 3.**

**Hybrid Structural Methods** leverage known individual dynamics to compute the expectation over next states and actions, but sample common noise.

Access to individual dynamics means that the mean-field sequence is computed **analytically**:

$$\boldsymbol{\mu}_{t+1} = \mathbf{A}^{\pi}_{\boldsymbol{\mu}_t, z_t}{}^{\top} \boldsymbol{\mu}_t. \tag{8}$$

The policy is evaluated using the exact expectation over next individual state-transitions, but Monte Carlo rollouts for the common noise:

$$\hat{V}(s, \boldsymbol{\mu}_t, z_t) = \mathbb{E}_{a \sim \pi(\cdot | s, \boldsymbol{\mu}_t, z_t)} \left[ R(s, a, \boldsymbol{\mu}_t, z_t) + \gamma \mathbb{E}_{s' \sim \mathcal{T}(\cdot | s, a, \boldsymbol{\mu}_t, z_t)} \left[ \hat{V}(s', \boldsymbol{\mu}_{t+1}, z_{t+1}) \right] \right], \tag{9}$$

where $z_{t+1} \sim \Xi(\cdot \mid z_t)$. In state-vectorised form:

$$\hat{\mathbf{v}}_{\boldsymbol{\mu}_t, z_t} = \left[ \boldsymbol{\Pi}_{\boldsymbol{\mu}_t, z_t} \odot \mathbf{R}_{\boldsymbol{\mu}_t, z_t} \right] \mathbf{1} + \gamma \mathbf{A}^{\pi}_{\boldsymbol{\mu}_t, z_t} \left[ \hat{\mathbf{v}}_{\boldsymbol{\mu}_{t+1}, z_{t+1}} \right], \tag{10}$$

where the variables are defined as in Information Box 1.

**Analytic Mean-Field Update in Partially-Observable Settings**

$$\mu_{t+1}(s_{t+1}) := p(s_{t+1} \mid \tilde{\mu}_t, z_t) = \int_{\mathcal{H}_t} \int_{\mathcal{A}} \mathcal{T}(s_{t+1} \mid s_t, a_t, \mu_t, z_t) \pi(a_t \mid \tau_t) \tilde{\mu}_t(\tau_t) \mathrm{d}a_t \mathrm{d}\tau_t, \tag{11}$$

$$\tilde{\mu}_{t+1}(\tau_{t+1}) := p(\tau_{t+1} \mid \tilde{\mu}_t, z_t, z_{t+1}) = \mathcal{U}(o_{t+1} \mid s_{t+1}, \mu_{t+1}, z_{t+1}) \mathcal{T}(s_{t+1} \mid s_t, a_t, \mu_t, z_t) \pi(a_t \mid \tau_t) \tilde{\mu}_t(\tau_t), \tag{12}$$

Under Assumptions 1.1 and 6.1, the control problem for any exogenous mean-field behaviour admits an optimal policy which only conditions on $(s_t, o_{0:t})$. This means that we restrict our search for a Nash equilibrium to policies only conditioning on $(s_t, o_{0:t})$.

The main benefit is that, because agents condition on the shared public observation history $o_{0:t}$, the mean-field remains a distribution over the fixed state space $\mathcal{S}$ rather than over the exponentially growing history space $\mathcal{H}_t$, as demonstrated by Equation 13. Although the length of shared observation history $o_{0:t}$ grows linearly with time, in practice, it is not stored explicitly, but encoded in the hidden state of a recurrent neural network. In contrast, conditioning on IAOHs would require distinct hidden states for an exponential number of histories.

**Practical Implementation.** Figure 1 (right) illustrates the RSPG architecture. The hidden state remains independent of the individual state because only observations of the aggregate state are processed recurrently. This means that all agents share the same recurrent embedding at each timestep. In practice, we compute this embedding once per timestep and broadcast it to all states, so that the additional cost of RSPG over SPG is a single GRU update per timestep, independent of the number of states.

For continuous actions, we parameterise an underlying continuous distribution and construct a categorical policy by evaluating its density on a fixed action grid. To preserve the ordinal structure of the action space, we apply a softmax transformation to the corresponding log-densities to obtain categorical probabilities. We find that such a parameterisation outperforms directly learning categorical logits; ablations are provided in Appendix D.4.1.

Pseudocode is provided in Algorithm 1 in Appendix B.1. At each iteration, we sample $E$ environments in parallel, rollout the endogenous mean-field sequence, and compute discounted returns backward through time. Gradients propagate through individual state transitions and expected rewards, but not through mean-field transitions.

## 7  MFAX

**Environment Framework.** To evaluate RSPG and enable further research into more complex MFG problem settings, we now introduce MFAX, our JAX based MFG framework. Unlike existing libraries, MFAX separates environments with white-box access to individual transition dynamics from those with black-box access, supporting both analytic and sample-based mean-field updates. Also unlike existing libraries, MFAX accommodates partial observability, common noise, and multiple initial mean-field distributions.

Since in most applications the transition kernel has sparse support, MFAX evaluates matrix-vector products (e.g. Equations 8 and 10 in Information Box 3) in functional form.

**Analytic Mean-Field Update in Partially-Observable Settings with Public Information**

$$\mu_{t+1}(s_{t+1}) := p(s_{t+1} \mid o_{0:t}, \mu_t, z_t) = \int_{\mathcal{S}} \int_{\mathcal{A}} \mathcal{T}(s_{t+1} \mid s_t, a_t, \mu_t, z_t)\pi(a_t \mid s_t, o_{0:t})\mu_t(s_t)\mathrm{d}a_t\mathrm{d}s_t. \qquad (13)$$

*Table 1.* Comparison of **Open-Source MFG libraries**.

| | MEAN-FIELD UPDATE | | TRANSITION DYNAMICS | | FEATURES | | |
|---|---|---|---|---|---|---|---|
| LIBRARY | ANALYTIC | SAMPLE | WHITE-BOX | BLACK-BOX | PART. OBSERVABLE | COMMON NOISE | MULTIPLE $\mu_0$ |
| MFAX | Y | Y | Y | Y | Y | Y | Y |
| OPENSPIEL | Y | N | N | Y | N | N | N |
| MFGLIB | Y | N | N | Y | N | N | N |

*Table 2.* **Time for one mean-field update** of the Linear Quadratic environment (100 states and 7 actions (Wu et al., 2025)). Experiments averaged over 90 updates after 10 warm-up steps (excluding compilation time) and conducted on NVIDIA L40S GPUs (48 GB).

| LIBRARY | LANGUAGE | MEAN-FIELD UPDATE (SEC) |
|---|---|---|
| MFAX | PYTHON | ANALYTIC: $2.98 \times 10^{-4}$
SAMPLE: $4.35 \times 10^{-4}$ |
| OPENSPIEL | C++ | $5.44 \times 10^{-3}$ |
| MFGLIB | PYTHON | $3.58 \times 10^{-1}$ |

This avoids materialising the $|\mathcal{S}|^2$ transition matrix and reduces memory to the mean-field vector and policy matrix i.e. $O(|\mathcal{S}| + |\mathcal{S}||\mathcal{A}|)$. By parallelising across both states and actions, mean-field updates are also substantially faster than existing libraries. For example, updates of the Linear Quadratic environment (Laurière et al., 2022; Wu et al., 2025) are over $10\times$ and $1000\times$ faster than OpenSpiel (Lanctot et al., 2019) and MFGLib (Guo et al., 2023). Being JAX-based, MFAX also supports GPU-accelerated parallelism across environments.

In addition to environments, MFAX implements two HSM and three RL algorithms; further details are provided in Appendix C and our codebase.

**Environment Implementations.** Table 3 in Appendix C.4 summarises our test environments. **Linear Quadratic** and **Beach Bar**, two toy environments, are partially observable adaptations of standard MFG benchmarks with common noise (Perrin et al., 2020; Wu et al., 2025). In Linear Quadratic, agents are rewarded for concentrating together. In Beach Bar, agents are rewarded for remaining near the bar while it is open, but penalised for being next to it at closure, which can randomly occur halfway through the episode. In both environments, agents observe the mean state, but not the timestep, common noise realisation, or full mean-field distribution. Beach Bar therefore tests whether anticipatory behaviour has been learned.

The **Macroeconomics** environment is a heterogeneous-agent model with common noise, where prices (interest rates and wages) are determined endogenously from the mean-field distribution (Krusell & Smith, 1998). The two-dimensional individual state represents wealth and income, the one-dimensional action the proportion of wealth consumed and the one-dimensional common noise the aggregate productivity shock. Agents must learn to maximise discounted utility from consumption, balancing immediate reward with saving for the future. Following Moll (2026), agents observe prices, but not the timestep, common noise or full mean-field distribution. Additional details are provided in Appendix C.

## 8 Experiments

Our experiments test whether (i) RSPG's rate of convergence remains as fast as SPG's and (ii) RSPG learns history-aware behaviour. We compare HSMs against RL-based methods to highlight that, by treating transition dynamics as a black box, RL-based methods lead to slower convergence to their final exploitability than HSMs.

**Ablations and Benchmarks.** We ablate RSPG with its history-independent counterpart, SPG, which we extend to use a multilayer perceptron (MLP) rather than a tabular policy. We also benchmark HSMs with purely sample-based RL algorithms: Independent PPO (IPPO) (Schulman et al., 2017; Algumaei et al., 2023; De Witt et al., 2020), Recurrent IPPO (Ni et al., 2021) and M-OMD (Laurière et al., 2023). We do not include AFP, as prior work (Lauriere et al., 2022; Perolat et al., 2021) shows that M-OMD converges significantly faster and avoids repeated best-response computation. We refer the reader to Appendix C.2 for more details.

**Evaluation Metrics.** Exploitability (Laurière et al., 2022) quantifies the maximum increase in expected return that a player can achieve by deviating from the policy used by the rest of the population: $\mathcal{X}_{\Pi_{\text{eval}}}(\pi) =$

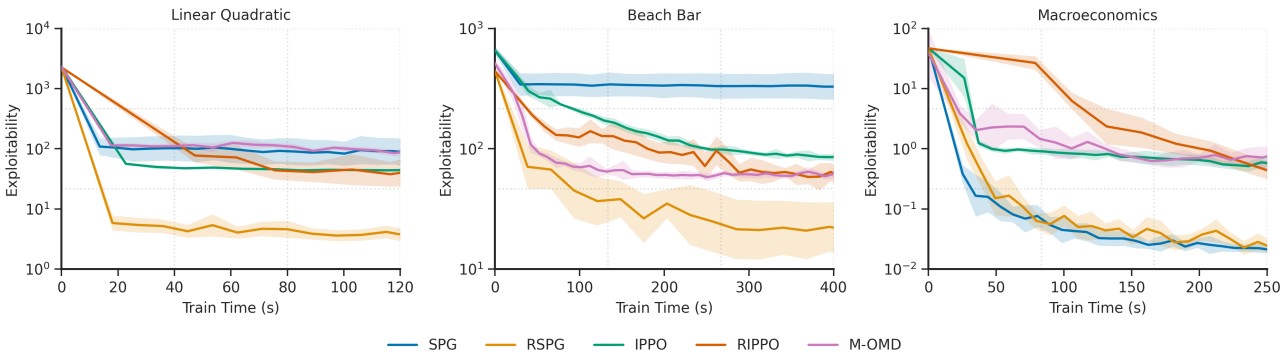

*Figure 2.* **Exploitability** versus **training wall-clock time** for partially observable **Linear Quadratic**, **Beach Bar**, and **Macroeconomics** environments. Experiments conducted on NVIDIA L40S GPUs (48 GB). SPG and RSPG are HSMs; IPPO, RIPPO and M-OMD are RL-based methods. 95% confidence intervals over 10 seeds.

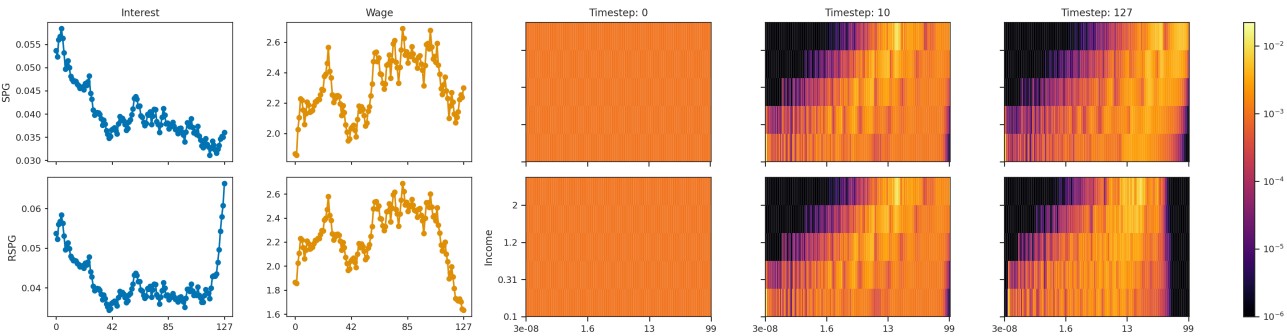

*Figure 3.* **Heatmaps: mean-field** distribution (**income** on y-axis and **wealth** on x-axis) at specific timesteps during the episode for **Macroeconomics** environment (total episode length of 128 steps). **Interest rates** (**first column**) and **wages** (**second column**) are determined by the mean-field distribution.

$\mathbb{E}\left[\sup_{\pi' \in \Pi_{\text{eval}}} J\left(\pi', \pi\right) - J\left(\pi, \pi\right)\right]$, where the expectation is taken over the initial aggregate state and common-noise trajectory. When $\Pi_{\text{eval}}$ equals the admissible policy class, exploitability measures the proximity to a Nash equilibrium. In our experiments, to enable fair comparison between memoryless and history-aware methods, we use a larger perfect-information class $\Pi_{\text{PI}}$ for evaluation, so $\mathcal{X}_{\Pi_{\text{PI}}}$ is a conservative regret-like upper bound rather than an exact distance to equilibrium. We approximate exploitability by averaging over sampled common noise sequences and compute $\sup_{\pi' \in \Pi_{\text{PI}}} J\left(\pi', \pi\right)$ via backward induction since all environments admit exact analytic mean-field updates.

Because HSMs track states rather than agents, comparing algorithms by environment steps is not meaningful. We therefore compare methods using wall-clock training time. To ensure reproducibility, we release all implementations and hyperparameter configurations (see Appendix D.2). Finally, we visualise the evolution of the mean-field and learned policy to identify interesting behaviour.

**Results.** Our experiments demonstrate that RSPG captures history-dependence without compromising proximity to an

MFG Nash equilibrium; in fact, RSPG often achieves a lower exploitability at a faster rate than its benchmarks.

Figure 2 shows that RSPG is the only method that consistently achieves competitive exploitability across all environments, converging to the lowest or second-lowest exploitability. SPG performs comparably to RSPG in the Macroeconomics environment because the observation provides substantial information about the aggregate state, reducing the importance of memory. However, as discussed below, RSPG identifies a more behaviourally-realistic equilibrium. M-OMD, SPG, and IPPO perform poorly in the Beach Bar and Linear Quadratic environments because they learn memoryless policies.

Notably, despite the Macroeconomics environment being nearly fully observable, M-OMD underperforms. We hypothesise this to be due to the environment's larger state-action space, where Q-function based methods struggle to exploit the underlying ordinal structure of the action space. This is not the case for policy-based methods, where we can more naturally parameterise an underlying continuous distribution (Section 6 and Figure 10).

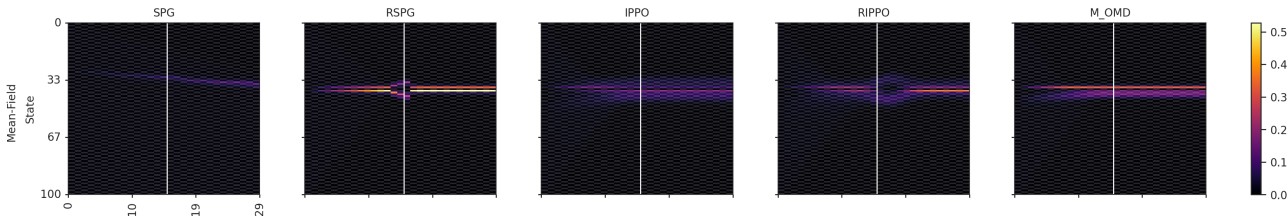

*Figure 4.* **Mean-field** distribution (y-axis) versus time (x-axis) for the **Beach Bar** environment. Agents are rewarded for being next to the bar when it is open, and penalised for being directly next to the bar at closure time, which can occur halfway through the episode (white-line).

As anticipated, SPG and RSPG achieve approximately an order-of-magnitude faster convergence than RL-based methods. HSMs learn directly from mean-field rollouts, while RL methods incur additional overhead from single-agent rollouts between successive mean-field rollouts. RIPPO is slower than IPPO and M-OMD because maintaining temporal consistency for history-awareness requires us to sequentially recalculate the loss, in contrast to recalculating the loss in parallel for memoryless algorithms.

Visualising the learned mean-field distributions shows that, unlike memoryless agents, history-dependent agents acquire anticipatory behaviour. This is particularly evident in the Beach Bar and Macroeconomics environments (Figures 4 and 3). With RSPG and RIPPO, agents learn to move away from the Beach Bar just before the potential closure time. Similarly, in the Macroeconomics environment, agents learn to spend their wealth near the end of the episode, which pushes wages down and interest rates up. Interest rates are determined by the mean-field distribution, and negatively correlated with percentage of agents in high wealth states. Even though SPG and RSPG attain similar levels of exploitability in the Macroeconomics environment (Figure 2), only RSPG captures anticipatory behaviour; in this sense, RSPG identifies a more behaviourally realistic equilibrium without sacrificing exploitability. We refer the reader to Appendix D.1 for more visualisations.

When understanding Figure 2, note that reduced variance in policy gradients, as provided by HSMs, does not necessarily translate to lower variance in final performance. Variability in exploitability across seeds can arise from non-convex optimisation, different initialisations, and convergence to different equilibria. We additionally note that HSMs still sample common noise.

## 9   Conclusion

In this work we present RSPG, the first history-aware HSM leveraging known transition dynamics for lower variance updates. Empirically, we show that RSPG achieves an order-of-magnitude faster convergence than sample-based RL methods, and more realistic agent behaviour than memoryless

SPG. We also present MFAX, our JAX-based environment framework for MFGs. Unlike other MFG libraries, MFAX distinguishes between mean-field environments with white-box and black-box access to the transition model, as well as supporting partial observability, common noise and multiple initial distributions.

## 10   Future Work

In the future, HSMs could be applied to more complex settings involving games with major and minor players or multiple mean-fields, where the common noise may include the behaviour of the major player or opposing mean-fields. HSM variance reduction could be incorporated into generalised advantage estimation within an actor-critic framework rather than differentiating through the full rollout.

MFAX enables the implementation of more complex environments, such as ones with individual agent observation functions or threshold-based reward functions, which could be used to simulate phenomena such as bank runs, for example. More generally, the library could incorporate a third mean-field update wrapper that supports function approximation for the analytic mean-field update, thereby allowing extensions to higher-dimensional individual state and action spaces. For example, we could use an attention-based network to learn the analytic mean-field update from real-world data while focusing only on volatile regions of the mean-field.

Finally, one limitation of our work is that we do not provide convergence guarantees to an MFG Nash equilibrium. Extending guarantees to our more complex problem settings is non-trivial; existing ones rely on strong assumptions (e.g. full-observability, monotonicity, and continuous reward and transition functions (Hu & Zhang, 2025; Perrin et al., 2020; Perolat et al., 2021; Cui & Koeppl, 2021)), which do not hold in our applications. Yongacoglu et al. (2024) establish convergence for independent learning in partially observable MFGs with shared aggregate observations, but only for memoryless policies. We remark that despite no convergence guarantees, we observed stable convergence for RSPG across all experiments.

## Impact Statement

This paper presents work whose goal is to advance the field of machine learning. There are many potential societal consequences of our work, none of which we feel must be specifically highlighted here.

## Acknowledgements

The authors would like to thank **Michael Beukman**, **Nathan Monette** and **Darius Muglich** for their helpful advice and comments on the project.

**Clarisse Wibault** is funded by EPSRC grant EP/W524311/1. **Johannes Forkel** is funded by UKRI grant EP/Y028481/1. **Juan Agustin Duque** is supported by the St-Pierre-Larochelle Scholarship at the University of Montreal and by Aaron Courville's CIFAR AI Chair in Representations that Generalize Systematically. **George Whittle** is funded by EPSRC grant EP/W524311/1. **Yucheng Yang** is funded by Swiss NSF grant 10003091. **Chiyuan Wang** is supported by NSFC 72450002. **Benjamin Moll** is partially funded by ERC grant 101200645. **Jakob Foerster** is partially funded by the UKRI grant EP/Y028481/1, as well as being supported by the JPMC and Amazon Research Awards.

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

# A    Counter-Example for Assumption 6.1

We provide a simple example of a POMFG-CN with public observations in which the IAOH $\tau_t$ contains information about the aggregate state that is not contained in the current individual state $s_t$ and public observation history $o_{0:t}$.

Consider a POMFG-CN with no non-trivial actions, state space $\mathcal{S} = \{0, 1\}$ and common-noise space $\mathcal{Z} = \{0, 1\}$. Let the common noise be persistent such that $z_{t+1} = z_t$ and $p_{z_0}(0) = p_{z_0}(1) = \frac{1}{2}$.

Let the initial mean-field be deterministic and uniform such that $\mu_0(0) = \mu_0(1) = \frac{1}{2}$.

Suppose that the shared public observations are uninformative, i.e. $\mathcal{O} = \{\perp\}$ and $o_t = \perp$ for all $t$.

The individual state transition depends on the common noise as follows:

$$s_{t+1} = \begin{cases} s_t, & z_t = 0, \\ 1 - s_t, & z_t = 1. \end{cases}$$

Since $\mu_t$ is uniform, the next mean-field is also uniform under either common-noise realisation, i.e. $\mu_{t+1}(0) = \mu_{t+1}(1) = \frac{1}{2}$.

Consider the belief in $z_t$ given only the current state $s_t$ and the public observation history $o_{0:t}$ at time $t = 1$. Since $o_{0:1}$ is uninformative and we do not observe $s_0$, the current state alone and public history do not reveal the common noise.

By contrast, the full individual history $\tau_1 = (s_0, o_0, s_1, o_1)$ reveals the common noise. Explicitly, $s_1 = s_0$ implies that $z_1 = 0$, whereas $s_1 = 1 - s_0$ implies that $z_1 = 1$. For example, if $\tau_1 = (0, \perp, 0, \perp)$, then $p(z_1 = 0 \mid \tau_1) = 1$ whereas $p(z_1 = 0 \mid s_1 = 0, o_{0:1}) = \frac{1}{2}$.

Since $\mu_1$ is deterministic and uniform, this implies that $p(\mu_1, z_1 \mid \tau_1) \neq p(\mu_1, z_1 \mid s_1, o_{0:1})$. Therefore, $p(\mu_t, z_t \mid \tau_t) = p(\mu_t, z_t \mid s_t, o_{0:t})$ does not hold trivially. The example shows that an agent's private trajectory can contain information about the aggregate state beyond the current individual state and the public observation history.

To make the informational difference payoff-relevant, suppose that at time $t = 1$ the agent chooses an action $a_1 \in \{0, 1\}$ and receives a reward $R(s_1, a_1, \mu_1, z_1) = \mathbf{1}\{a_1 = z_1\}$. A policy conditioning on the full history $\tau_1$ can infer $z_1$ perfectly and obtain an expected reward of 1, whereas a policy conditioning only on $(s_1, o_{0:1})$ cannot distinguish the two common-noise realisations and obtains an expected reward of 1/2. In such an example, restricting policies to $\pi(\cdot \mid s_t, o_{0:t})$ is strictly suboptimal.

# B  Methods

## B.1  Recurrent Structural Policy Gradient

We provide the pseudocode for RSPG below. Gradients are allowed to propagate through the expected rewards over actions and the individual state transitions, but not through the mean-field updates or environment reward function. $d_t$ is a reset mask used to reset recurrent states across episodes.

---

**Algorithm 1** RSPG

---

**input:** parameters $\theta$
**repeat**
  **for** $E$ environments in parallel **do**
    $\mu_0 \sim p_{\mu_0}, \ z_0 \sim p_{z_0}, \ o_0 = \mathcal{U}(\mu_0, z_0), \ h_0 \leftarrow \mathbf{0}, \ d_0 \leftarrow 0, \mathbf{v}_H \leftarrow \mathbf{0} \quad \text{or} \quad \mathbf{v}_H[s] \leftarrow R_H(s, \boldsymbol{\mu}_H, z_H)$
    **for** $t = 0 : H - 1$ **do**                          $\triangleright$ Generate mean-field sequence using analytic mean-field update
      $h_{t+1} \leftarrow \mathrm{GRU}_\theta(h_t, o_t, d_t)$
      $\mathbf{\Pi}_t[s, a] \leftarrow \pi_\theta(a \mid s, h_{t+1})$                              $\triangleright$ For every state-action pair
      $\boldsymbol{\mu}_{t+1} \leftarrow \text{stop-gradient}\left( \mathbf{A}_{\mu_t, z_t, o_{0:t}}^{\pi}{}^\top \boldsymbol{\mu}_t \right)$                   $\triangleright$ Environment transition
      $\mathbf{R}_t[s, a] \leftarrow \text{stop-gradient}\left( R(\mathbf{s}, a, \boldsymbol{\mu}_t, z_t) \right), \quad z_{t+1} \sim \Xi(\cdot \mid z_t), \quad o_{t+1} = \mathcal{U}(\mu_{t+1}, z_{t+1})$
    **end for**                                         $\triangleright$ Return $\{\mu_t, z_t, o_{0:t}\}_{0:H}, \{\mathbf{\Pi}_t, \mathbf{R}_t\}_{0:H-1}$
    **for** $t = H - 1 : 0$ steps **do**                 $\triangleright$ Calculate discounted return backwards through time
      $\mathbf{v}_t \leftarrow (\mathbf{\Pi}_t \odot \mathbf{R}_t) \cdot \mathbf{1} + \gamma \cdot \mathbf{A}_{\mu_t, z_t, o_{0:t}}^{\pi} \mathbf{v}_{t+1}$
    **end for**                                                  $\triangleright$ Return $\mathbf{v}_0$
  **end for**                                                  $\triangleright$ Return $\{\boldsymbol{\mu}_0, \mathbf{v}_0\}_{1:E}$
  $J \leftarrow \frac{1}{E} \sum \boldsymbol{\mu}_0 \cdot \mathbf{v}_0$
  $\theta \leftarrow \theta + \alpha_\pi \nabla J$                                $\triangleright$ Update policy parameters
**until** convergence

---

# C  Benchmark Structure

JAX-based open-source implementations inspiring MFAX include PureJaxRL (Lu et al., 2022), Gymnax (Lange, 2022) and JaxMARL (Rutherford et al., 2024), respectively implementing single file RL algorithms; single-agent RL environments; and multi-agent environments and algorithms.

## C.1  MFAX Environment Structure

In this section, we provide a brief overview of MFAX's modular environment structure.

### C.1.1  BASE ENVIRONMENTS

Environments are required to have a base class that implements *deterministic* single state step and reward functions —which return the deterministic next individual states (i.e. $T(s_t, a_t, \mu_t, z_t)$; we use $T$ rather than $\mathcal{T}$ to show that the transition is deterministic) and rewards (i.e. $R(s_t, a_t, \mu_t, z_t)$) given the current individual state, aggregate state and a single action— and functions determining when the environment sequence has terminated or truncated.

### C.1.2  ANALYTIC MEAN-FIELD WRAPPER

Broadly speaking, analytic mean-field environments must implement:

1. a single state step function, which adds idiosyncratic noise to the deterministic next state (e.g. $s_{t+1} = T(s_t, a_t, \mu_t, z_t) + \epsilon_t$, where $\epsilon_t$ might be normally distributed noise) and returns the next state indices as well as their probabilities;

2. an aggregate observation function $\mathcal{U}(\mu_t, z_t)$, which returns the common observation of the aggregate state;

3. and a reset function, which determines the exact initial mean-field distribution $\mu_0 \sim p_{\mu_0}$.

Given these functions, the generic environment analytic mean-field update wrapper handles updating the entire mean-field distribution (which involves computing $\mathbf{A}^\top \boldsymbol{\mu}$), calculating expectations over next states (such as $\mathbf{Av}$, for example) and returning the reward matrix $\mathbf{R}$.

Rather than constructing or storing $\mathbf{A}$, we implement the associated matrix–vector products (the analytic mean-field update and expectation) in functional form, with the mean-field distribution, policy, vector to be pre-multiplied by, and common noise as inputs. This avoids materialising the transition matrices altogether, reducing memory requirements from $|\mathcal{S}|^2$ to $O(|\mathcal{S}| + |\mathcal{S}||\mathcal{A}|)$, which is linear in the number of states. This is much more efficient in practice, since in many applications (such as two-dimensional physical spaces and macroeconomic models) the transition kernel $\mathcal{T}(s' \mid s, a, \mu, z)$ is sparse, inducing a sparse transition matrix $\mathbf{A}$, while the action space is much smaller than the state space ($|\mathcal{A}| \ll |\mathcal{S}|$).

**Mean-field sequence**  The pseudocode for generating the mean-field sequence for an analytic mean-field environment is presented in Algorithm 2; "analytic mean-field" represents the functional form of pre-multiplying by the $\mathbf{A}^\top$ matrix.

---

**Algorithm 2** Mean-field sequence generation for analytic mean-field environment

---

  **input:** (Recurrent) policy $\pi$
  $\mu_0 \sim p_{\mu_0}, \ z_0 \sim p_{z_0}, \ o_0 = \mathcal{U}(\mu_0, z_0), \ (h_0 \leftarrow \mathbf{0}), \ d_0 \leftarrow 0$
  **for** $t = 0 : H - 1$ **do**
     $(\boldsymbol{\Pi}_t[s, a], h_{t+1}) \leftarrow \pi(\cdot \mid \mathbf{s}, o_t, h_t, d_t)$   (or   $\boldsymbol{\Pi}_t[s, a] \leftarrow \pi(\cdot \mid \mathbf{s}, o_t)$   if non-recurrent)
     $\boldsymbol{\mu}_{t+1} \leftarrow$ analytic-mean-field-update$(\boldsymbol{\mu}_t, z_t, \boldsymbol{\Pi}_t)$                     ▷ Environment transition
     $\mathbf{R}_t[s, a] \leftarrow R(\mathbf{s}, a, \boldsymbol{\mu}_t, z_t), \quad z_{t+1} \sim \Xi(\cdot \mid z_t), \quad o_{t+1} \sim \mathcal{U}(\mu_{t+1}, z_{t+1})$
  **end for**                                                        ▷ Return $\{\mu_t, z_t\}_{0:H}$

---

### C.1.3  SAMPLE-BASED WRAPPER

Broadly speaking, sample-based mean-field environments must implement:

1. a single agent step function, which steps a single agent forward given the current individual state, aggregate state and

action by sampling idiosyncratic noise and adding that to the deterministic next state from the base environment (i.e. $s_{t+1} = T(s_t, a_t, \mu_t, z_t) + \epsilon_t$);

2. a local observation function, which returns the individual observation of the aggregate state (i.e. $o_t \sim \mathcal{U}(\cdot \mid s_t, \mu_t, z_t)$);

3. a reset function, which samples individual agents from an initial mean-field distribution (i.e. $s_0 \sim \mu_0$ with $\mu_0 \sim p_{\mu_0}$).

Given these functions, the generic environment sample-based wrapper handles updating the entire mean-field distribution by stepping a fixed number of agents forwards and re-computing the mean-field statistics based on the updated samples.

**Mean-field sequence** The pseudocode for generating the mean-field sequence for a sample-based environment is presented in Algorithm 3.

---

**Algorithm 3** Mean-field sequence generation for sample-based environment

---

**input:** (Recurrent) policy $\pi$
$\mu_0 \sim p_{\mu_0}$, $z_0 \sim p_{z_0}$, $(o_0 \leftarrow \mathcal{U}(\mu_0, z_0))$, $(h_0 \leftarrow \mathbf{0})$, $(d_0 \leftarrow 0)$
**for** $N$ agents in parallel **do**
    $s_{i,0} \sim \mu_0$
    $(o_{i,0} \sim \mathcal{U}(\cdot \mid s_{i,0}, \mu_0, z_0))$          $\triangleright$ (Rather than shared observation if observations are individual-state-dependent).
**end for**
**for** $t = 0 : H - 1$ **do**
    **for** $N$ agents in parallel **do**
        $(\pi_t, h_{i,t+1}) \leftarrow \pi(\cdot \mid s_{i,t}, o_{i,t}, h_{i,t}, d_t)$   (or   $\pi_t \leftarrow \pi(\cdot \mid s_{i,t}, o_{i,t})$   if non-recurrent)
        $a_{i,t} \sim \pi_t, \quad s_{i,t+1} \sim \mathcal{T}(\cdot \mid s_{i,t}, a_{i,t}, \mu_t, z_t), \quad r_{i,t+1} = R(s_{i,t}, a_{i,t}, \mu_t, z_t)$
    **end for**
    $z_{t+1} \sim \Xi(\cdot \mid z_t), \quad \mu_{t+1} \leftarrow \frac{1}{N} \sum_i \delta_{s_{i,t+1}}, \quad (o_{t+1} \leftarrow \mathcal{U}(\mu_{t+1}, z_{t+1}))$
    **for** $N$ agents in parallel **do**
        $(o_{i,t+1} \sim \mathcal{U}(\cdot \mid s_{i,t+1}, \mu_{t+1}, z_{t+1}))$
    **end for**
**end for**                    $\triangleright$ Return $\{\mu_t, z_t\}_{0:H}$

---

## C.2 MFAX Algorithms

We implement the following algorithms in MFAX:

**SPG** We implement SPG (Yang et al., 2025) using an MLP rather than a tabular policy. SPG is implemented in the same way as RSPG (pseudocode given in Appendix B.1), but without history.

**RSPG** We implement RSPG (pseudocode given in Appendix B.1).

**IPPO** We implement a mean-field version of Independent Proximal Policy Optimization (Schulman et al., 2017; De Witt et al., 2020; Algumaei et al., 2023) as, despite its theoretical shortcomings, it has been shown to outperform joint learning approaches in various Multi-Agent Reinforcement Learning tasks due to its robustness to environment non-stationarity (De Witt et al., 2020). We provide the pseudocode for our implementation below.

---

**Algorithm 4** IPPO

---

**input:** Policy $\pi$ with parameters $\theta$
**repeat**
    **for** $E$ environments in parallel **do**
        $\triangleright$ Generate mean-field sequence using sample-based mean-field distribution
    **end for**                  $\triangleright$ Return $\{\mu_{0:H}, z_{0:H}\}_{1:E}$
    $\triangleright$ Update $\theta$ using single-agent PPO for N agents within each environment.
**until** convergence

---

**RIPPO** We implement a mean-field version of Recurrent IPPO (Ni et al., 2021). The pseudocode is identical to that of IPPO, but the policy includes an RNN.

**M-OMD** We implement M-OMD, an algorithm specifically designed for MFGs, which amends the DQN target to $y = r(s, a) + \alpha\tau \log \pi(a \mid s) + \gamma \mathbb{E}_{a' \sim \pi(\cdot \mid s')} [Q(s', a') - \tau \log \pi(a' \mid s')]$ and uses a softmax rather than argmax policy to smooth policy updates, while still allowing for the policy to become greedier with time. Modifying the reward rather than the optimisation objective amends the convergence trajectory of $(\pi, \mu_{0:H})$ without altering the final fixed-point converged to. M-OMD and its variations (Wu et al., 2025; Benjamin & Abate) are state-of-the-art for most MFG problem settings. The pseudocode is given in (Wu et al., 2025).

## C.3 MFAX Evaluation Metrics

To evaluate the algorithms, MFAX computes an approximate exploitability using backwards induction, if environments are implemented using the analytic mean-field update wrapper. For both analytic mean-field and sample-based environment wrappers, MFAX computes the per-agent mean-expected return for that environment. All algorithms allow the training wall-clock time to be measured.

## C.4 MFAX Specific Environment Details

*Table 3.* Overview and comparison of the features and spatial dimensions of the Linear Quadratic, Beach Bar and Macroeconomics test environments.

| ENVIRONMENT | FEATURES | | | SPATIAL DIMENSIONS | | |
|---|---|---|---|---|---|---|
| | PARTIALLY OBSERVABLE | COMMON NOISE | MULTIPLE $\mu_0$ | $\lvert\mathcal{S}\rvert$ | $\lvert\mathcal{A}\rvert$ | $\lvert\mathcal{Z}\rvert$ |
| LINEAR QUADRATIC | Y | Y | N | 100 | 7 | 2 |
| BEACH BAR | Y | Y | Y | 100 | 11 | 2 |
| MACROECONOMICS | Y | Y | N | 1000 | $\infty$ | $\infty$ |

### C.4.1 LINEAR QUADRATIC ENVIRONMENT

The first environment we implement is a partially observable adaptation of the Linear Quadratic environment with common noise that is standard in MFG literature (Perrin et al., 2020; Wu et al., 2025). The environment is finite horizon, with a fixed initial mean-field distribution. The individual state space, the state space of the common noise and the action space are all 1D and given by:

$$\mathcal{S} = \{0, 1, 2, \cdots, 99\}, \quad \mathcal{Z} = \{-1, 1\}, \quad \mathcal{A} = \{-3, -2, -1, 0, 1, 2, 3\}.$$

The initial mean-field and common noise are given by:

$$\mu_0 = \text{Unif}(\mathcal{S}), \quad z_0 \sim \text{Unif}(\{-1, 1\}).$$

The transition dynamics are given by:

$$s_{t+1} = \text{clip}\left(\text{round}\left(s_t + a_t + \sigma \cdot \left(\xi_t \cdot \rho + \epsilon \cdot \sqrt{1 - \rho^2}\right)\right), 0, \lvert\mathcal{S}\rvert - 1\right), \quad z_{t+1} = z_t,$$

where $\xi_t$ has a piecewise effect depending on the instantiation of the common noise $z_t$:

$$\xi_t = \begin{cases} -10 \, z_t, & t < 8, \\ 0 \, z_t, & 8 \leq t \leq 20, \\ 10 \, z_t, & t > 20, \end{cases}$$

and the idiosyncratic noise $\epsilon \sim \mathcal{N}(0, 1)$, is, in practice, discretised on a finite support (e.g. $\{-3, ..., 3\}$) with appropriately normalised probabilities.

The reward is given by:

$$r_t = -c_a \cdot a_t^2 + q \cdot a_t \cdot [\boldsymbol{\mu}_t \cdot \boldsymbol{s} - s_t] - \frac{\kappa}{2} [\boldsymbol{\mu}_t \cdot \boldsymbol{s} - s_t]^2 \quad \text{if} \quad t \neq H, \quad r_H = -\frac{c_{term}}{2} [\boldsymbol{\mu}_t \cdot \boldsymbol{s} - s_t]^2,$$

effectively encouraging agents to concentrate together. The observation of the aggregate state $g = (\boldsymbol{\mu}, z, t)$ is 1D and given by:

$$o_t = \boldsymbol{\mu}_t \cdot \boldsymbol{s},$$

where $\boldsymbol{s}$ denotes a vector of length $|\mathcal{S}|$ containing the index of state, such that the observation is just the mean state of the mean-field. We use the following parameters for the implementation:

*Table 4.* Parameters used for the Linear-Quadratic environment.

| PARAMETER | DESCRIPTION | VALUE |
|---|---|---|
| $\sigma$ | IDIOSYNCRATIC NOISE STD | 1.0 |
| $\rho$ | NOISE PARAMETER | 0.5 |
| $c_a$ | REWARD ACTION COST | 0.5 |
| $c_{\text{TERM}}$ | REWARD TERMINAL COST | 1.0 |
| $q$ | REWARD ACTION DIRECTION PARAMETER | 0.1 |
| $\kappa$ | REWARD DISTANCE PENALTY | 0.5 |
| $H$ | TIME HORIZON | 30 |
| $|\mathcal{S}|$ | NUMBER OF STATES | 100 |
| $|\mathcal{A}|$ | NUMBER OF ACTIONS | 7 |
| $|\mathcal{Z}|$ | NUMBER OF COMMON NOISE REALISATIONS | 2 |

### C.4.2 BEACH BAR 1D ENVIRONMENT

The second environment we implement is a partially observable version of the beach-bar environment, again standard in MFG literature (Perrin et al., 2020; Wu et al., 2025). The environment is finite horizon. The aggregate state is given by $g = (\boldsymbol{\mu}, z, l, t)$, where $l$ is the location of the bar that is set at the beginning of the episode and hence independent of time, and $z$, the common noise, determines whether the bar closes at $t = H/2$. The individual state space, the state space of the common noise and the action space are all 1D and given by:

$$\mathcal{S} = \{0, 1, 2, \cdots, 99\}, \quad \mathcal{Z} = \{0, 1\}, \quad \mathcal{A} = \{-5, -4, -3, -2, -1, 0, 1, 2, 3, 4, 5\}.$$

The initial mean-field and common noise are given by:

$$l_0 \sim \text{Unif}(\mathcal{S}), \quad \mu_0 = \text{Unif}(\mathcal{S} \backslash \{l_0\}), \quad z_0 \sim \text{Unif}(\{0, 1\}).$$

The transition dynamics are given by:

$$s_{t+1} = \text{legal}\left[s_t + a_t + \epsilon\right], \quad z_{t+1} = z_t,$$

where $\epsilon$ is an instantiation of the idiosyncratic noise and is sampled from $\{-2, -1, 0, 1, 2\}$ with probability $\{0.05, 0.1, 0.7, 0.1, 0.05\}$, and "legal" is a function that prevents agents from moving into illegal states. The instantiation of the common noise $\xi$ is given by:

$$\xi_t = \begin{cases} 1, & t < H/2, \\ z_t, & t \geq H/2. \end{cases}$$

The reward is given by:

$$\begin{aligned}
r_t = & - |l - s_t|\, \xi_t \\
& - |\mathcal{S}|\, (1 - z_t)\, \mathbf{1}\{t \geq \tfrac{H}{2} - 1\}\, \mathbf{1}\{|l - s_t| = 1\} \\
& - \xi_t\, \text{clip}(\log \boldsymbol{\mu}_t[s_t], -|\mathcal{S}|, 0) \\
& - 0.1\, |\mathcal{S}|\, (1 - \xi_t)\, \text{clip}(\log \boldsymbol{\mu}_t[s_t], -|\mathcal{S}|, 0) \\
& - \frac{|a_t|}{|\mathcal{S}|}.
\end{aligned}$$

where the first term penalises being far from the bar when it is open, the second term is a strong penalty for being next to the bar when the bar is closed, the third term is a weak reward for being in sparse areas when the bar is open, and the fourth term is a stronger reward for being in sparse areas when the bar is closed. The observation is 3D and given by:

$$o_t = (\boldsymbol{\mu}_t \cdot \boldsymbol{s}, \xi_t, l).$$

again, $\boldsymbol{s}$ denotes a vector of length $|\mathcal{S}|$ containing the index of state, such that the first element of the observation tuple is just the mean state of the mean-field. The observation does not include the timestep, meaning that a notion of time can only be encoded in the history.

*Table 5.* Parameters used for the Beach-Bar environment.

| PARAMETER | DESCRIPTION | VALUE |
|---|---|---|
| $H$ | TIME HORIZON | 30 |
| $|\mathcal{S}|$ | NUMBER OF STATES | 100 |
| $|\mathcal{A}|$ | NUMBER OF ACTIONS | 11 |
| $|\mathcal{Z}|$ | NUMBER OF COMMON NOISE REALISATIONS | 2 |

### C.4.3 MACROECONOMICS

The Macroeconomics environment we implement is a heterogeneous agent model with common noise developed by Krusell & Smith (1998), where prices are endogenously determined from the mean-field distribution. The environment is long, but finite horizon, with a fixed initial mean-field distribution.

The individual state space is two-dimensional—with interpretation of wealth and income—and the action space is one-dimensional, representing the proportion of its budget constraint that an agent chooses to consume.

$$\mathcal{S} = [0, 99] \times [0.1, 2.0] \subset \mathbb{R}^2, \quad \mathcal{Z} = \mathbb{R}, \quad \mathcal{A} = [0, 1] \subset \mathbb{R}.$$

In practice, to track the mean-field distribution, we discretise the continuous state space $\mathcal{S}$ on a finite grid $\mathcal{S}^h = \mathcal{S}_1^h \times \mathcal{S}_2^h$ with $|\mathcal{S}_1^h| \times |\mathcal{S}_2^h|$ points, obtained via a geometric discretisation of each state dimension. The mean-field distribution $\boldsymbol{\mu}$ is represented on $\mathcal{S}^h$, while individual state dynamics are defined on the continuous space $\mathcal{S}$. To evaluate the analytic mean-field update, we linearly discretise the action space $\mathcal{A}$.

The initial mean-field and common noise are given by:

$$\mu_0 = \mathrm{Unif}(\mathcal{S}), \quad z_0 = 0.$$

For clarity, in the following we use a dash to denote time $t + 1$ and subscripts to denote the state component, since the state is two dimensional. The transition dynamics $\mathcal{T}(s' \mid s, a, \boldsymbol{\mu}, z)$ are given by

$$s_1' = \mathrm{clip}\big((1 - a)\big((1 + p_1^*(\boldsymbol{\mu}, z))s_1 + p_2^*(\boldsymbol{\mu}, z)\, s_2\big), 0, 99\big),$$
$$s_1^{h'} := D_1^h(s_1'),$$
$$s_2^{h'} = \begin{cases} s_2^{i-1}, & \text{with probability } 0.1, \\ s_2^i, & \text{with probability } 0.8, \\ s_2^{i+1}, & \text{with probability } 0.1, \end{cases}$$
$$s_2' := D_2^{h^{-1}}(s_2^{h'})$$

where $s_2^i$ is the current grid point of the second state component (income), and $s_1$ represents the continuous value of the first component (wealth). $D$ is the discretisation function. Transitions are clipped at the boundary of $\mathcal{S}_2^h$.

The common noise evolves as:

$$z' = \rho_z \cdot z + \nu_z \cdot \epsilon, \quad \epsilon \sim \mathcal{N}(0, 1). \tag{14}$$

The quantities $p_1^*$ and $p_2^*$ have the interpretation of the interest rate and wage, and are defined via the following price functionals:

$$p_1^*(\boldsymbol{\mu}, z) = \frac{\partial}{\partial \bar{s}_1(\boldsymbol{\mu})} F(\boldsymbol{\mu}, z), \quad p_2^*(\boldsymbol{\mu}, z) = \frac{\partial}{\partial \bar{s}_2(\boldsymbol{\mu})} F(\boldsymbol{\mu}, z),$$

where $F$ is the production function of a representative firm:

$$F(\boldsymbol{\mu}, z) = \exp(z)\, \bar{s}_1^{\alpha}\, \bar{s}_2^{1-\alpha}, \quad \bar{s}_k(\boldsymbol{\mu}) = \int_{\mathcal{S}} s_k\, d\boldsymbol{\mu}(s) = \mathbb{E}_{(s_1, s_2) \sim \boldsymbol{\mu}}[s_k],\ k \in \{1, 2\}.$$

The reward is given by:

$$r = \frac{\left(a \cdot ((1 + p_1^*(\boldsymbol{\mu}, z))s_1 + p_2^*(\boldsymbol{\mu}, z)s_2)\right)^{1-\sigma}}{1 - \sigma}.$$

Following Moll (2026), we consider a partially observable case in which the observation is two-dimensional and given by:

$$o = (p_1^*(\boldsymbol{\mu}, z),\ p_2^*(\boldsymbol{\mu}, z)),$$

i.e., in addition to their individual states $s$, agents observe the two prices $p_1^*$ and $p_2^*$ but not the entire distribution $\boldsymbol{\mu}$.

We use the following parameters for the implementation:

*Table 6.* Parameters used for the Macroeconomics environment.

| PARAMETER | DESCRIPTION | VALUE |
|---|---|---|
| $\alpha$ | COBB-DOUGLAS: CAPITAL SHARE | 0.36 |
| $\gamma$ | DISCOUNT FACTOR | 0.95 |
| $\sigma$ | COEFFICIENT OF RELATIVE RISK AVERSION | 2 |
| $\rho_z$ | PERSISTENCE OF AR(1) FOR $z$ | 0.9 |
| $\nu_z$ | VOLATILITY OF AR(1) FOR $z$ | 0.03 |
| $H$ | TIME HORIZON | 128 |
| $\mathcal{S}_1$ | STATE SPACE FOR $\mathcal{S}_1$ | [0, 99] |
| $\mathcal{S}_2$ | STATE SPACE FOR $\mathcal{S}_2$ | [0.1, 2] |
| $\mathcal{A}$ | ACTION SPACE | [0, 1] |
| $|\mathcal{S}_1^h|$ | DISCRETISED NUMBER OF STATES $\mathcal{S}_1$ | 200 |
| $|\mathcal{S}_2^h|$ | DISCRETISED NUMBER OF STATES $\mathcal{S}_2$ | 5 |
| $|\mathcal{A}^h|$ | DISCRETISED NUMBER OF ACTIONS | 20 |

# D   Experiments

## D.1   Further Results

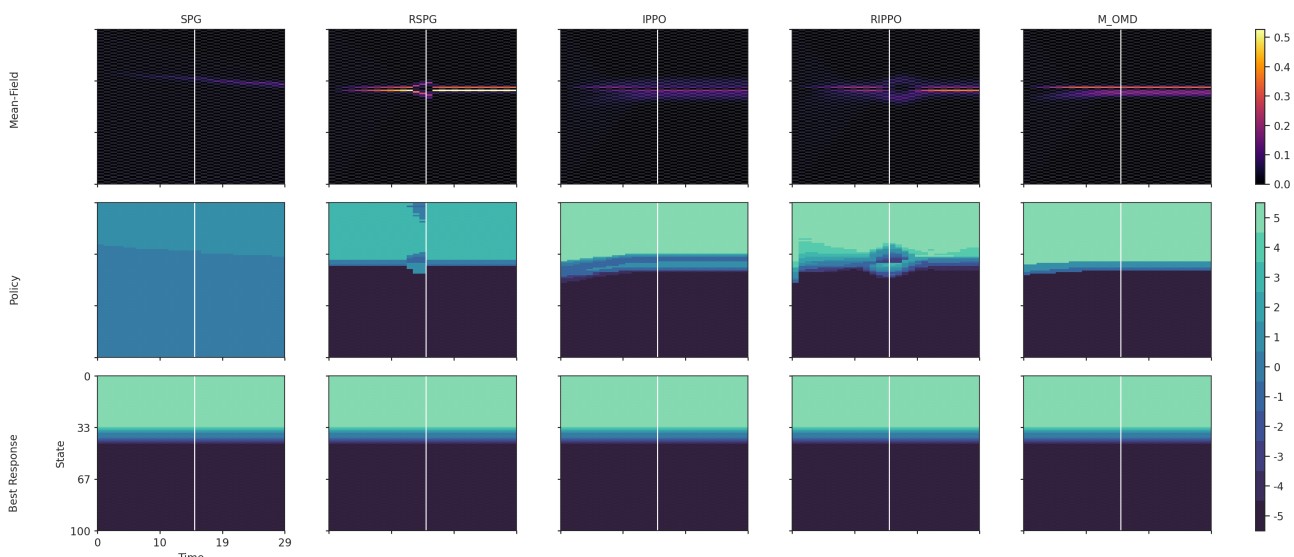

*Figure 5.* **Mean-field** distribution (y-axis) versus time (x-axis) for the **Beach Bar** environment (**top**). **Learned policy** (**middle**) versus **best response policy** (**bottom**). Agents are rewarded for being next to the bar when it is open, and penalised for being directly next to the bar when it is closed, or just before it closes, which can occur halfway through the episode (white-line). Here the bar stays open, which is why agents move back towards the bar.

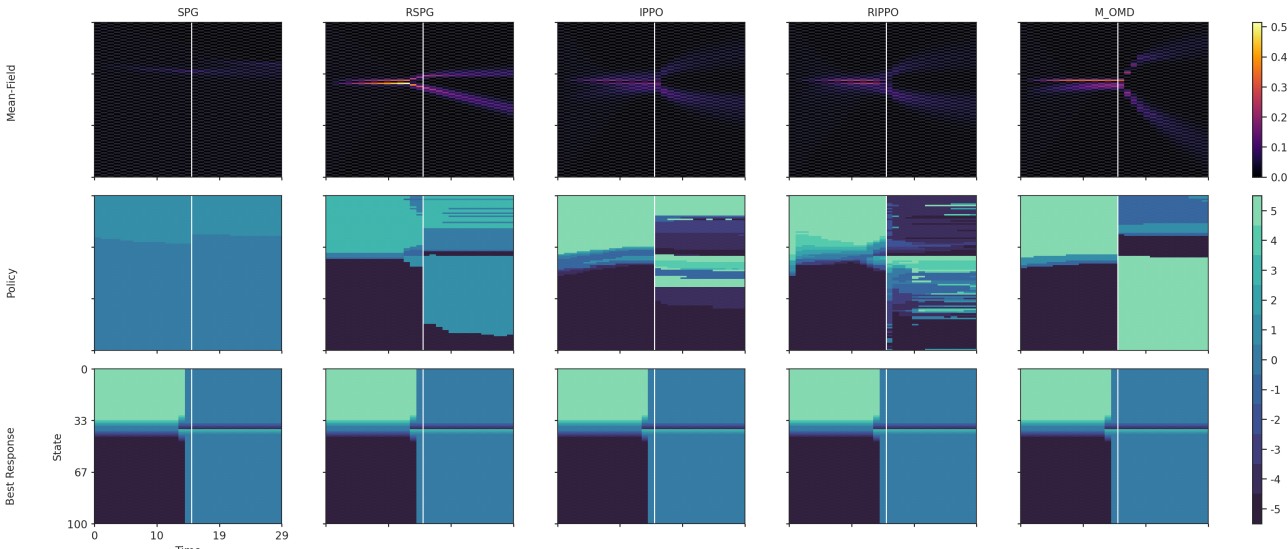

*Figure 6.* **Mean-field** distribution (y-axis) versus time (x-axis) for the **Beach Bar** environment (**top**). **Learned policy** (**middle**) versus **best response policy** (**bottom**). Agents are rewarded for being next to the bar when it is open, and penalised for being directly next to the bar when it is closed, or just before it closes, which can occur halfway through the episode (white-line). Here the bar stays closed, which is why agents move away from the bar.

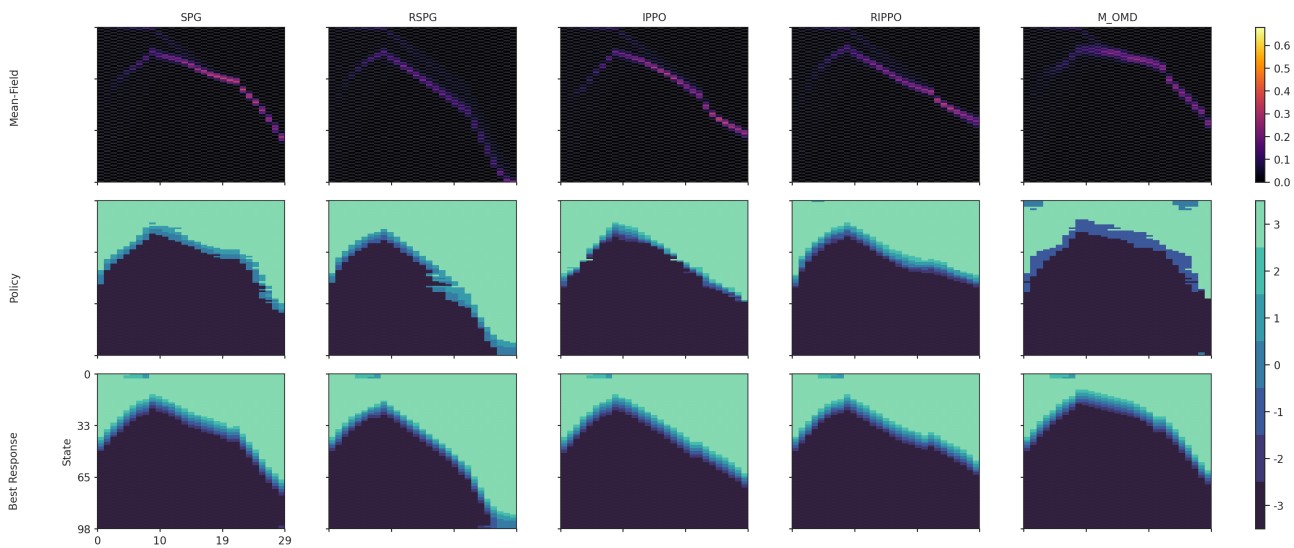

*Figure 7.* **Mean-field** distribution (y-axis) versus time (x-axis) for the **Linear Quadratic** environment (**top**). **Learned policy** (**middle**) versus **best response policy** (**bottom**). Agents are subject to one of two realisations of common noise, pushing the entire population downwards or upwards.

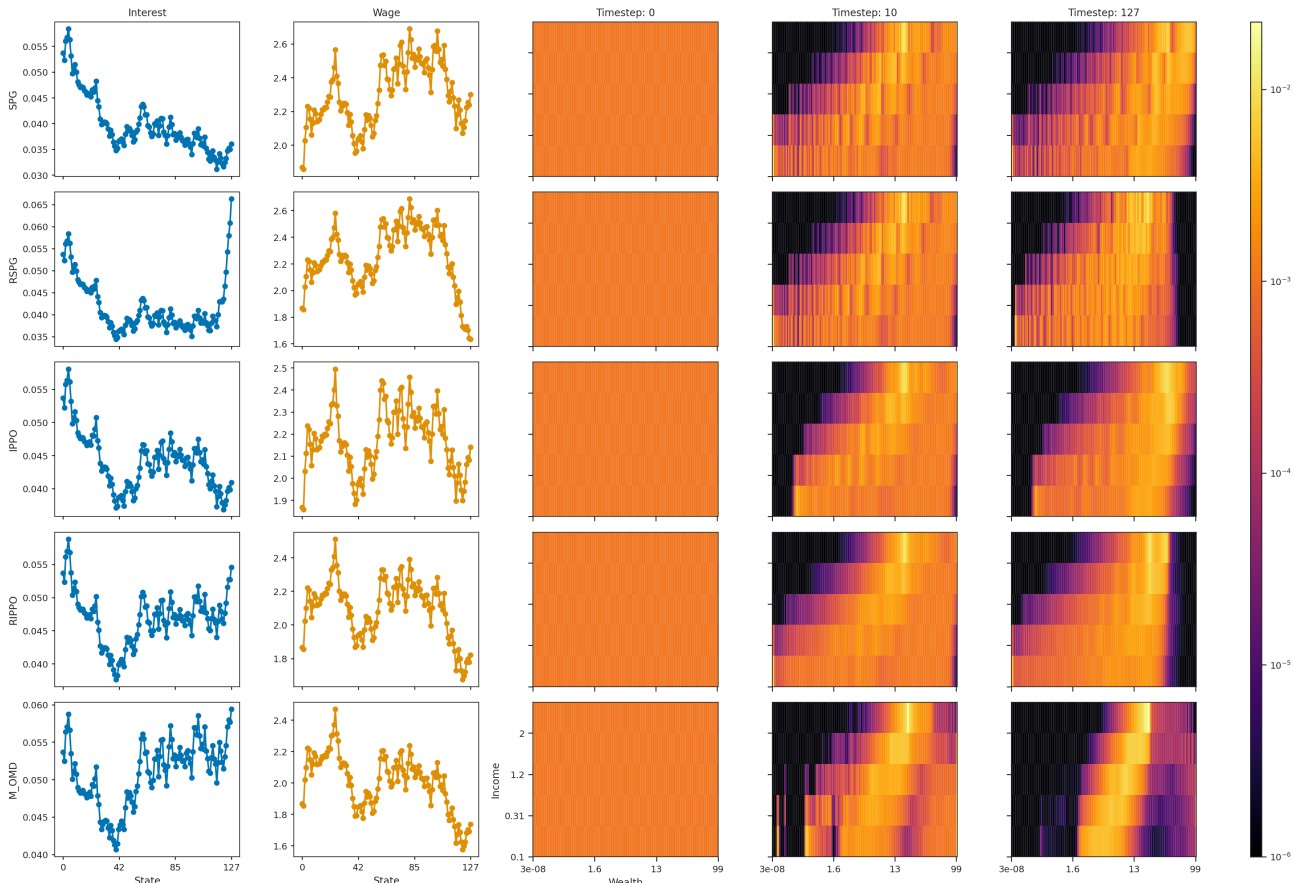

*Figure 8.* **Heatmaps: mean-field** distribution (**income** on y-axis and **wealth** on x-axis) at specific timesteps during the episode for the **Macroeconomics** environment (with total episode length of 128 steps). **Interest rates** (**first column**) and **wages** (**second column**) are determined by the mean-field distribution. The environment is implemented as a finite horizon: with RSPG and RIPPO, we see that agents learn anticipatory behaviour, spending more wealth just before the end of the episode, pushing interest rates up and wages down. This is not the case for the memoryless policies.

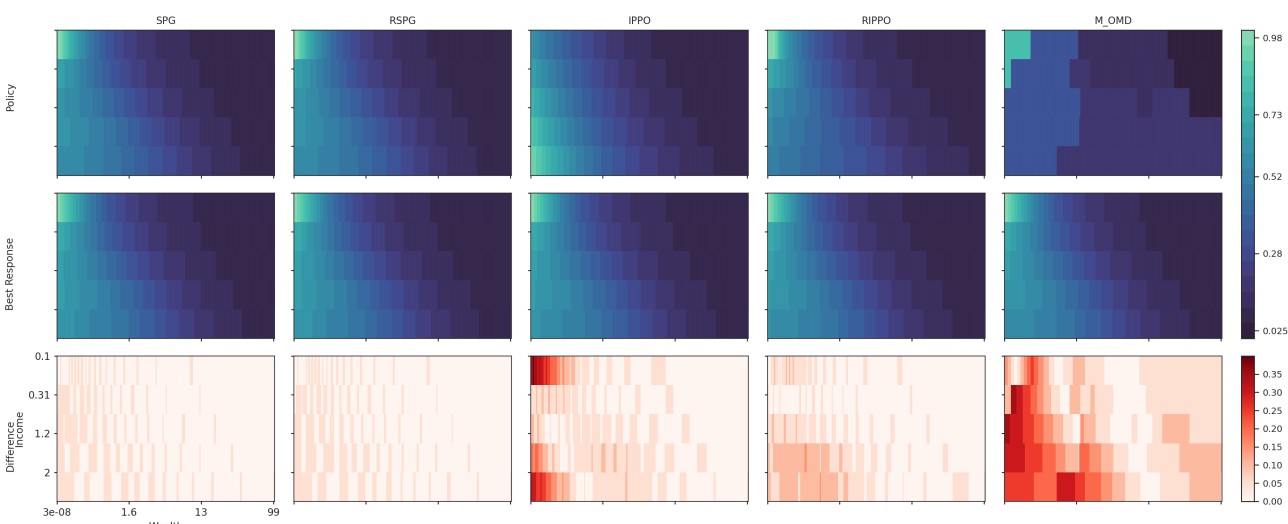

*Figure 9.* **Learned actions** (**top**) and **best response actions** (**middle**) and (**difference**) between the two (**bottom**) for the **Macroeconomics** environment. For M-OMD, the learned policy is much coarser, which we attribute to the fact that it is not parameterised by an underlying continuous action distribution.

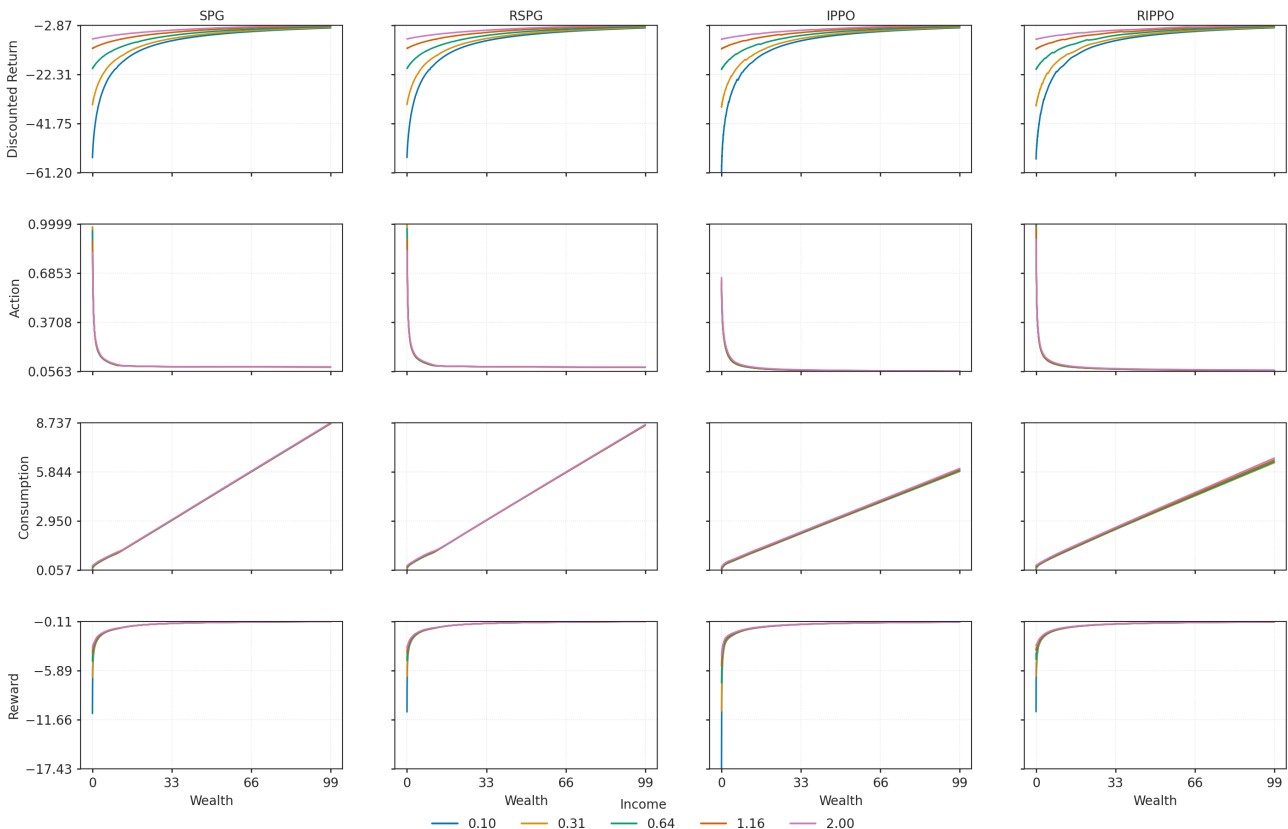

*Figure 10.* Underlying continuous **actions** (**second row**), and their associated continuous **consumptions** (**third row**), **rewards** (**final row**) and **cumulative discounted rewards** (**top row**) over the entire episode for the **Macroeconomics** environment. We include these plots for macroeconomics interest. M-OMD is omitted, since it does not have an underlying continuous action distribution.

## D.2 Implementation Details and Hyperparameters

**Network Structures** We use the following architectures.

For environments with state indices as inputs, we use an embedding layer of size 64 to encode the index. Otherwise, we use a feed-forward layer of hidden size 64.

For history-aware policies, observations are encoded using a GRU with hidden size 64, and then passed through one layer of ReLU activation and a feed-forward layer of hidden size 64.

For memoryless policies, observations are encoded using a feed forward layer of hidden size 64.

Encoded states and observations are then concatenated and passed through an MLP with 3 feed forward layers with hidden size 128 and ReLU activation.

Policy networks used for RSPG, SPG, IPPO and RIPPO have categorical heads for the toy environments and continuous Beta distribution heads for the Macroeconomics environment.

We use Adam for all training and anneal the learning rate using a linear schedule, reducing the learning rate by a factor of 10 over training.

**Hyperparameter Sweeps** We use the same memory- and budget-related hyperparameters (such as number of parallelised environments) as well as capacity-related hyperparameters (such as network architecture) for all algorithms. For algorithm-specific hyperparameters, we perform sweeps over identical ranges when applicable, and report results using configurations that achieve both low exploitability and fast convergence. SPG and RSPG have relatively few hyperparameters, and we therefore sweep only over the learning rate. For the RL-based algorithms, in addition to the learning rate, we sweep over the number of updates between mean-field sequence rollouts, which significantly affects wall-clock training time. SPG and RSPG do not have single-agent rollouts between mean-field sequence rollouts, which significantly reduces the training time. For the RL algorithms, we also sweep over the number of single agents per parallelised environment. For IPPO and RIPPO we additionally sweep over the entropy coefficient, and the amended DQN target parameters $\tau$ and $\alpha$ for M-OMD. In total, we sweep over 3 hyperparameter configurations for SPG and RSPG, 81 for IPPO and RIPPO, and 243 for M-OMD.

*Table 7.* Hyperparameters used for our algorithmic implementations. Remaining hyperparameters were swept over (see Table 8).

| | HSMs | | RL | | |
| --- | --- | --- | --- | --- | --- |
| HYPERPARAMETER | SPG | RSPG | IPPO | RIPPO | M-OMD |
| MAX GRAD NORM | 1.0 | 1.0 | 1.0 | 1.0 | 1.0 |
| NUM STEPS | – | – | 64 | 64 | – |
| NUM EPOCHS | – | – | 1 | 1 | – |
| NUM MINIBATCHES | – | – | 8 | 8 | – |
| GAE LAMBDA | – | – | 0.95 | 0.95 | – |
| CLIP EPSILON | – | – | 0.2 | 0.2 | – |
| VF COEF | – | – | 0.5 | 0.5 | – |
| LEARN EVERY | – | – | – | – | 8 |
| LOSS | – | – | – | – | MSE |
| BUFFER CAPACITY | – | – | – | – | 300000 |
| BATCH SIZE | – | – | – | – | 2048 |
| MIN BUFFER SIZE TO LEARN | – | – | – | – | 10000 |
| UPDATE TARGET NET | – | – | – | – | 512 |
| MIN BUFFER STEPS | – | – | – | – | 1000 |
| EPSILON DECAY DURATION (PERCENTAGE) | – | – | – | – | 0.5 |
| EPSILON START | – | – | – | – | 1.0 |
| EPSILON END | – | – | – | – | 0.1 |
| RESET REPLAY BUFFER ON ITERATION | – | – | – | – | TRUE |

*Table 8.* Hyperparameter Sweep Values.

| HYPERPARAMETER | SWEEP VALUES |
| --- | --- |
| LEARNING RATE | [0.0001, 0.001, 0.01] |
| NUMBER OF UPDATES PER MEAN FIELD SEQUENCE ROLLOUT STEP | [50, 100, 200] |
| NUMBER OF SINGLE AGENTS PER ENVIRONMENT | [8, 128, 1024] |
| ENTROPY COEF | [0.001, 0.01, 0.1] |
| TAU | [0.05, 5, 10] |
| ALPHA | [0.9, 0.95, 0.99] |

We use a higher number of parallelised environments for Beach Bar, since, unlike Linear Quadratic and Macroeconomics, it has multiple initial distributions.

*Table 9.* Number of Parallelised Environments (same for all algorithms).

| ENVIRONMENT | NUMBER OF PARALLELISED ENVIRONMENTS |
| --- | --- |
| LINEAR QUADRATIC | 8 |
| BEACH BAR | 128 |
| MACROECONOMICS | 8 |

**Environments**  For our sample-based environments, we use $10,000$ agents to approximate the mean-field. We ensure that the number of agents is much larger than the individual state-space $N \gg |\mathcal{S}|$ such that this is not a limiting factor in algorithmic performance. Since agents are v-mapped, this does not add to training time.

## D.3  Compute

Unless stated otherwise, all experiments were run on NVIDIA L40S GPUs (48 GB).

## D.4  Ablations

### D.4.1  POLICY REPRESENTATION

The policy parameterisation significantly influences the final exploitability. HSMs that compute the exact analytic mean-field update require integration over the policy. For continuous action spaces (such as those encountered in macroeconomic environments) we approximate this integral by discretising the policy: the continuous action distribution is evaluated at uniformly spaced points over the action domain and then renormalised. This discretisation consistently outperforms a categorical parameterisation, which lacks an inductive bias reflecting the ordinal structure of the action space, and consistently converges to a sub-optimal solution.

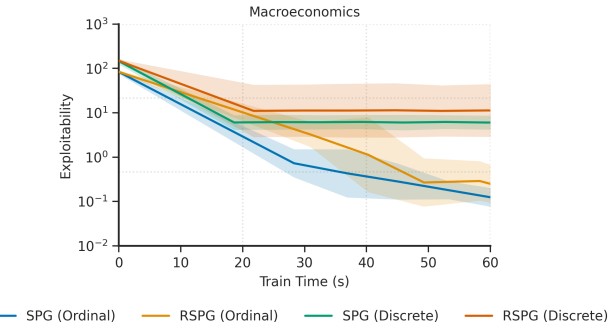

*Figure 11.* **Exploitability** versus **training wall-clock time** for partially observable **Macroeconomics** environment with ordinal versus categorical policies. Experiments were run on NVIDIA A40 GPUs. Having an underlying continuous distribution (**Ordinal**) consistently outperforms the underlying categorical distribution (**Discrete**), which does not encode information about the ordinal nature of the action space.

