# OpenReview forum: "Recurrent Structural Policy Gradient for Partially Observable Mean Field Games"
_ICML.cc/2026/Conference — ICML 2026 spotlight_

### Official Review · Reviewer_WxGU · 2026-02-23

**Soundness:** 4
**Presentation:** 3
**Significance:** 3
**Originality:** 3
**Overall Recommendation:** 5
**Confidence:** 3

**Summary:**

This paper tackles the problem of finding approximate equilibria in partially observable mean field games. The authors build on the existing literature on Hybrid Structural Methods and add a history/memory component to existing HSMs. After some theoretical results, the paper evaluate the proposed RL-based learning approach against different existing algorithms for various problem instances. Apart from that, the authors formulate a JAX-based learning environment to drastically speed up computations compared to existing libraries like OpenSpiel and MFGLib.

**Compliance With Llm Reviewing Policy:**

Affirmed.

**Final Justification:**

The authors have addressed my questions adequately.

**Key Questions For Authors:**

Maybe I missed it, but is there an immediate conceptual connection between RSPG and MFAX? Or did the authors "just" develop MFAX because they required a computationally fast framework for RSPG?

**Limitations:**

yes

**Strengths And Weaknesses:**

I enjoyed reading the paper and did not spot any major weaknesses; here are the main strengths:
- The ideas are explained and developed in a conceptually clear fashion
- The comprehensive empirical evaluation yields convincing results for the proposed learning approach
- The empirical observations are supplemented by some theoretical insights
- The authors also outline in which scenarios their approach might not be the ideal choice
- Finally, I think that a JAX-based library for MFGs is an important tool for many future works on learning MFGs because it enables significantly faster computation times for different areas in MFG research.

---

> ### Author Rebuttal · Authors · 2026-03-28
>
> We thank the reviewer for their positive response to our work.
>
> In response to their key question, although it is faster, MFAX is not strictly required for HSMs (such as SPG or RSPG), since RL algorithms also require updating the mean-field distribution (but involve additional, agent-based sampling between mean-field updates), and current MFG libraries implement exact mean-field updates (which is possible with low-dimensional individual state-spaces). However, MFAX was required to validate learning history-aware behaviour with RSPG, since current libraries do not support partially observable environments.
>
> MFAX was mainly developed for speed, ease-of-use, and to enable future research in MFGs. We wanted a library that clearly distinguishes different MFG problem settings (standard RL problem settings, with potentially high-dimensional individual state-spaces, from modelling applications with low-dimensional ones (our applications of interest in this paper)). This is why, unlike current MFG libraries, MFAX enables implementation of both exact mean-field updates (required for HSMs i.e. differentiating through expected returns over next state transitions), and also sample-based mean-field updates (if an exact update were to be computationally intractable due to a high dimensional individual state-space, for example).
>
> Please let us know if there is anything that we can clarify further.
>
>
> **References**
>
> Note that references are only abbreviated here due to space constraints, but are exact in the updated work.
>
> [1] Moll, B. The Trouble with Rational Expectations in Heterogeneous Agent Models: A Challenge for Macroeconomics, 2025.
>
> [2] Cardaliaguet et al. Learning in Mean Field Games: the Fictitious Play, July 2015.
>
> [3] Elie et al. Contact rate epidemic control of COVID-19: an equilibrium view, 2020.
>
> [4] Alasseur et al. An Extended Mean Field Game for Storage in Smart Grids, April 2019.
>
> [5] Yang et al. Mean Field Game-Theoretic Framework for Interference and Energy-Aware Control in 5G Ultra-Dense Networks, February 2018.
>
> [6] Lauriere et al. Learning in Mean Field Games: A Survey, July 2024.
>
> [7] Achdou et al. Mean Field Games and Applications: Numerical Aspects, March 2020.
>
> [8] Lauriere, M. , Numerical Methods for Mean Field Games and Mean Field Type Control, June 2021.
>
> [9] Cardaliaguet et al., The master equation and the convergence problem in mean field games, September 2015.
>
> [10] Carmona, et al. Deep Learning for Mean Field Games and Mean Field Control with Applications to Finance,
> July 2021.
>
> [11] Hu et al. Recent Developments in Machine Learning Methods for Stochastic Control and Games, March 2024.
>
> [12] Perrin et al. Fictitious Play for Mean Field Games: Continuous Time Analysis and Applications, October 2020.
>
> [13] Han et al. A global solution method for heterogeneous agent models with aggregate shocks, 2022.
>
> [14] Yang, et al. Structural Reinforcement Learning for Heterogeneous Agent Macroeconomics, December 2025.
>
> [15] Perrin, S. Scaling up Multi-agent Reinforcement Learning with Mean Field Games and Vice-versa. 2022.
>
> [16] Algumaei et al. Regularization of the policy updates for stabilizing Mean Field Games, April 2023.
>
> [17] Hu et al. MF-OML: Online Mean-Field Reinforcement Learning with Occupation Measures for Large Population Games, September 2025.
>
> [18] Cui et al. Approximately Solving Mean Field Games via Entropy-Regularized Deep Reinforcement Learning,
> July 2022.
>
> [19] Wu et al. Population-aware Online Mirror Descent for Mean-Field Games with Common Noise by Deep Reinforcement Learning, September 2025.
>
> [20] Perrin et al. Mean Field Games Flock! The Reinforcement Learning Way, May 2021.
>
> [21] Subramanian et al. Partially Observable Mean Field Reinforcement Learning. 2021.
>
> [22] Benjamin et al. Improving Real-World Applicability of Networked Mean-Field Games using Function Approximation and Empirical Mean-Field Estimation. 2025.
>
> [23] Yongacoglu et al. Mean-Field Games With Finitely Many Players: Independent Learning and Subjectivity. 2024.
>
> [24] Saldi et al. Approximate Nash Equilibria in Partially Observed Stochastic Games with Mean-Field Interactions.  August 2019.
>
> [25] Hu et al., Learned Belief Search: Efficiently Improving Policies in Partially Observable Settings, 2021.
>
> [26] Perolat et al. Scaling up Mean Field Games with Online Mirror Descent, February 2021.
>
> [27] Lauriere et al., Scalable Deep Reinforcement Learning Algorithms for Mean Field Games, June 2022.

---

> > ### Author Rebuttal · Reviewer_WxGU · 2026-04-02
> >
> > I thank the authors for their detailed response. I am happy to keep my initial positive score.

---

### Official Review · Reviewer_soaF · 2026-03-11

**Soundness:** 3
**Presentation:** 3
**Significance:** 3
**Originality:** 3
**Overall Recommendation:** 5
**Confidence:** 3

**Summary:**

The paper studies partially observable mean-field games (PO-MFGs) with common noise, thereby addressing two layers of uncertainty: incomplete information and aggregate shocks that affect the entire population simultaneously. To avoid the intractability of fully history-dependent policies, the authors propose a shared-history architecture (“reduced policy”), which is embedded within a RSPG framework. This results in computational tractability comparable to memoryless policies. Combined with the HSM component, the approach yields lower wall-clock training times and improved performance in the presented experiments. The experiments are conducted using the authors’ proposed JAX-based simulator, MFAX, which provides accelerated mean-field updates and supports partial observability, common noise, and multiple initial mean-field distributions.

**Compliance With Llm Reviewing Policy:**

Affirmed.

**Final Justification:**

The authors have addressed the majority of my initial concerns.

**Key Questions For Authors:**

1. Lines 234–239: I am unclear why the reward function explicitly includes \mu’ and z’ as its inputs – as inputs. Should the reward not simply depend on the current mean-fields \mu and z (via equation 3)? The inputs to Equation (7) appear different from those described in lines 234–238. Could the authors clarify how these equations are connected and where the expressions in lines 234–238 are being used?

2. History of shared observations: Although the shared-history mechanism reduces complexity relative to full histories, the history still appears to grow with the state-space size and time horizon. Would this not still pose limitations for long-horizon problems?

3. Lines 253–256 and Appendix D.4: How is a continuous action distribution evaluated at a specific point? For a continuous distribution, the probability mass at a single point is zero. Some mathematical clarification of the discretization procedure would improve readability and avoid confusion.

4. Is there an explanation for the high variance observed in SPG performance? While RSPG appears convincing in the two toy examples, its performance in the macroeconomic example is less clear. Can we interpret RSPG as identifying a “better equilibrium”? Additionally, why were other recurrent MARL baselines not considered? For example, recurrent MA-PPO [1], though primarily used in cooperative MARL, could potentially be adapted as a stronger baseline than RIPPO. Similarly, classical MFG algorithms such as average fictitious play [2] could provide meaningful comparison points.

5. Is there any concrete experiment showcasing the claim of variance reduction by using HSMs?

6. Figure (3): “The environment is implemented as a finite horizon: with RSPG, agents learn anticipatory behavior, spending more wealth just before the end of the episode, pushing interest rates down and wages up.” However, this interpretation appears inconsistent with the visual trends in the figure. Could the authors clarify this discrepancy?

Minor questions/suggestions:

1. The paper introduces several ideas in tandem (RSPG, HSM, shared histories, etc.). While each idea has merit, it would help to more clearly articulate how each component individually contributes to simplifying learning in PO-MFGs. Currently, the ideas are somewhat interwoven (particularly in lines 239–247), making it difficult to isolate the novelty. A dedicated subsection explaining the precise combination of HSM + RSPG and their complementary roles would improve clarity.

2. Section 6.2 is somewhat difficult to interpret in its current form. It may be better presented as a remark, as the motivation and connection to the preceding discussion are not entirely clear. The paragraph may also give the impression that standard RL methods would be preferable to HSM; explicitly discussing issues such as variance reduction could clarify the intended message.

3. Sections C.1.2 and C.1.3 would benefit from additional mathematical expressions to support the stated claims.

4. An intuitive explanation of the best-response curve in Figure (5) would improve accessibility.

5. Lines 697–701 appear to contain a repeated sentence.

References

[1] Yu, Chao, et al. "The surprising effectiveness of ppo in cooperative multi-agent games." Advances in neural information processing systems 35 (2022): 24611-24624

[2] Lauriere, Mathieu, et al. "Scalable deep reinforcement learning algorithms for mean field games." International conference on machine learning. PMLR, 2022.

**Limitations:**

1. The history of shared observations still grows with size of state-space and with time making the approach unsuitable for longer horizons.

2. As noted by the authors, MFAX would benefit from more complex environments that more fully demonstrate the applicability of partially observable MFG methods to realistic large-population games.

3. Furthermore, as noted by the authors, their methodology heavily relies on having access to the individual transition dynamics and mean-field push-forward operator, which may not be available in many practical applications.

**Strengths And Weaknesses:**

Strengths

1. The paper addresses the timely and technically challenging problem of partially observable mean-field games with common noise, significantly extending the scope of standard MFG formulations.

2. The use of aggregate states to construct shared histories reduces memory requirements in PO-MFG settings. Proposition 1 provides theoretical grounding for this claim. Additionally, leveraging access to individual training dynamics also improves computational tractability.

3. MFAX appears to provide meaningful computational benefits, particularly in accelerating mean-field updates and enabling scalable experimentation.

Weaknesses

1. In its current form, the paper would benefit from improved presentation and organization. For instance, the transition from Section 6.2 to Section 7 appears somewhat abrupt. Certain mathematical steps (e.g., the discretization of policies in lines 253–256) would benefit from clearer formal expositions. Some concepts currently described in the appendix (see comments below) would also benefit from additional explanation.

2. While the simulation results demonstrate improvements over baselines in toy examples, they do not yet fully convince the reader of applicability in more complex, large-scale scenarios such as the macroeconomic setting considered in the paper.

---

> ### Author Rebuttal · Authors · 2026-03-28
>
> We thank the reviewer for their insightful comments, and address key points below. Please let us know if there is anything else that we can clarify. All references are included in our response to reviewer WxGU.
>
> **W1**
>
> We have improved clarity throughout, with particular emphasis on the introduction (assumptions and applications), related works (positioning), and methods.
>
> **W2**
>
> We refer the reviewer to our response to reviewer eZCK. In brief, qualitative analysis (Figure 3) in the macroeconomics environment validates that history-aware behaviour is learned with RSPG, but not with SPG, while not compromising proximity to equilibrium (Figure 2).
>
> **KQ1**
>
> We agree that the exposition was suboptimal in this part of the paper. To reduce confusion, we have revised our work to demonstrate intractability via the mean-field update (instead of the reward term). Let us nevertheless explain the previous notation: the expected reward at the next timestep (required to calculate the estimated return at the current timestep) depends on the updated mean-field, hence the inclusion of $\mu'$ and $z'$. Interpreting dashed variables as $t=1$ and undashed as $t=0$, the reward term in lines 234–238 corresponds to a single component (state $s_0$) of the vector resulting from $\mathbf{A}^\pi_{(\mu_0, z_0)} \tilde{\mathbf{r}}^\pi_{(\mu_1, z_1)}$. Unlike Equation (7), which conditions on observation histories, this example illustrates that computing such rewards under full history dependence requires integrating over the (conditional) distribution $p^\pi(\tau_t)$, which grows exponentially with time and is therefore intractable.
>
> **KQ2**
>
> While the history of shared observations grows linearly with time, in practice, this history is not stored explicitly, but encoded in the hidden state of an RNN, so memory remains constant with time. Since aggregate observations are shared, only a single hidden state is required. This contrasts with policies conditioning on individual histories (which require tracking separate hidden states per trajectory), where, since the number of trajectories grows exponentially with time, so does the memory requirement.
>
> **KQ3**
>
> In our implementation, we discretise the action space using a fixed grid, evaluate the policy’s PDF at each grid point, and use the log-densities as logits to define a categorical distribution via a softmax. This preserves the underlying ordinality of actions, which would not be enforced by directly learning categorical logits. We have clarified this procedure in our work.
>
> **KQ4**
>
> Note that reduced variance in policy gradients (as provided by HSMs) does not necessarily translate to lower variance in final performance. Variability in exploitability across seeds can arise from non-convex optimisation, different initialisations, and convergence to different equilibria. In addition, HSMs still sample common noise. In the macroeconomics environment, SPG and RSPG achieve similar exploitability, but RSPG captures history-dependent behaviour by learning anticipatory dynamics (increased spending toward the end of the episode), which SPG does not do. In this sense, RSPG identifies a better (more behaviourally realistic) equilibrium, without sacrificing exploitability. Regarding baselines, MAPPO is designed for cooperative settings; we instead use Independent PPO and its recurrent IPPO, which serve as the appropriate comparison in general-sum settings. We do not include Deep AFP, as prior work [27, 26] shows that Deep M-OMD (which we include) converges significantly faster and avoids repeated best-response computation.
>
> **KQ5**
>
> The fact that we have lower-variance gradient estimates follows directly from the mathematical definition of the return estimator (HSMs replace Monte Carlo sampling of next-state transitions with analytic expectations), though we also see this in faster convergence to final exploitability (Figure 2).
>
> **KQ6**
>
> We thank the reviewer for spotting the typo.
>
> **MQs**
>
> We have revised the paper to clearly distinguish HSMs, RSPG, and shared histories. In brief, HSMs leverage known transition dynamics to compute expectations over next-state transitions analytically, reducing variance relative to fully sample-based RL methods. SPG is a memoryless HSM; RSPG extends SPG to history-aware policies, while maintaining tractability via shared histories. We retain Section 6.2 to clarify when HSMs are appropriate, but have revised it for clarity. We have also added mathematical notation in Section C and an intuitive explanation of the best-response (the best possible action given that the rest of the population follows the learned policy).
>
> **Ls**
>
> To address limitation (1) and (3), we refer the reviewer to our response to KQ2 and to reviewer A6kJ. In response to (2), we note this as a direction for future research: current environments implemented in MFAX simply validate RSPG as a HSM that learns history-awareness, which is the core contribution of this work.

---

> > ### Author Rebuttal · Reviewer_soaF · 2026-04-03
> >
> > The authors have addressed my major concerns. I am happy to raise my original score

---

### Official Review · Reviewer_eZCK · 2026-03-13

**Soundness:** 3
**Presentation:** 2
**Significance:** 3
**Originality:** 2
**Overall Recommendation:** 5
**Confidence:** 3

**Summary:**

This paper introduces the Recurrent Structural Policy Gradient (RSPG), a Hybrid Structural Method (HSM) for Partially Observable Mean Field Games with Common Noise (POMFGs-CN). The core contribution is extending existing memoryless HSMs (specifically SPG) to history-aware policies by restricting agent memory to a shared public observation history rather than individual action-observation histories, which would otherwise render the structural expectation intractable. Experiments on three environments,  Linear Quadratic, Beach Bar, and a Krusell-Smith macroeconomics model, demonstrate that RSPG achieves lower or competitive exploitability compared to baselines and converges roughly an order of magnitude faster in wall-clock time than RL methods.

**Compliance With Llm Reviewing Policy:**

Affirmed.

**Final Justification:**

The rebuttal has addressed my main concern. I raised my score from 4 to 5

**Key Questions For Authors:**

1. Does RSPG still produce useful approximate policies when Assumption 1 is mildly violated? Does exploitability degrade smoothly or catastrophically?

2. The policy network encodes aggregate observations through a GRU while keeping the hidden state independent of the individual state. In practice, the hidden state at time t is a deterministic function of the shared observation history $\bar o_t$ under Assumption 1. Does this mean the GRU hidden state is the same for all agents at any given time step within an environment rollout? If so, can the recurrent computation be amortised across agents, and what is the actual computational overhead of RSPG relative to SPG on a per-update basis?

3. Can the author provide a discussion on the convergence of RSPG? While the paper notes that existing convergence results for MFG algorithms rely on assumptions like monotonicity that may not hold in practice, this does not excuse the absence of any discussion on that.

**Limitations:**

yes

**Strengths And Weaknesses:**

Strenghts:

1. **Theoretically clean**. The key insight,  that conditioning on shared rather than individual histories restores tractability of the structural expectation,  is interesting and precise. Proposition 1 and its proof (Appendix A) are rigorous, and Corollary 1 makes the policy-class reduction explicit. Prior HSMs are restricted to memoryless, tabular policies. Extending HSMs to history-aware policies is a non-trivial contribution: the paper is the first to show that variance-reduction from structural methods and history-awareness are simultaneously achievable.

2. **MFAX**: Providing a well-structured code is crucial for reproducibility. The 10× speedup over OpenSpiel C++ is substantial and the functional-form representation of the push-forward operator (reducing memory from O(|S|²) to O(|S| + |S||A| + |Z|)) is technically sound and practically important for discretised continuous-state models.

3. **Experiments**: Qualitative results support the narrative. Figures 3, 4, 5, and 6 convincingly demonstrate that history-aware agents (RSPG, RIPPO) learn genuinely anticipatory behaviour while memoryless agents (SPG, IPPO) do not. This qualitative validation goes beyond the exploitability curves and strengthens the argument for history-aware policies.
Another point: The authors use identical network capacities and budget constraints across all algorithms, sweep over algorithm-specific hyperparameters fairly, report 95% confidence intervals over 10 seeds, and compare on wall-clock time rather than environment steps. These are responsible experimental choices.

Weaknesses

1. Assumption 1 is severely restrictive. The entire tractability argument rests on the observation being independent of the individual state: $o \sim U(\cdot|\mu, z)$. This rules out any setting where agents receive observations that depend on their own position within the population. The paper lists these as motivating applications in the introduction but Assumption 1 disqualifies them. Section 6.2 acknowledges this trade-off but does so briefly and without quantifying how restrictive the assumption is across the cited application domains. The claim that this is "reasonable in applications such as finance or macroeconomics" is not defended.

2. **Asymmetry**. The comparison with RL baselines is not entirely fair. The paper compares wall-clock time, which is reasonable, but HSMs and RL methods operate under fundamentally different assumptions (white-box vs. black-box access to dynamics). Claiming an "order-of-magnitude faster convergence" without prominently caveating that HSMs have strictly more information is somewhat misleading. The contribution is really about showing that when white-box access is available, one should exploit it, which is less surprising.  The same asymmetry can be found in the exploitability metrics. The latter is computed using exact push-forward, which requires the same white-box access that HSMs assume. This is standard in MFG, but it increases the asymmetry in the evaluation between HSMs and RL methods.

3. **Partial Observability**. The macroeconomic environment is nearly fully observable, undermining the partial observability claim. The observation in the Krusell-Smith environment is the two prices $(p_1^*, p_2^*)$. These two scalars are deterministic functions of the mean-field distribution. The paper acknowledges that "SPG performs comparably to RSPG in the endogenous environment, likely because the observation provides substantial information about the aggregate state",  but states this as an observation rather than recognising it as a significant limitation of the chosen benchmark for evaluating partial observability.

---

> ### Author Rebuttal · Authors · 2026-03-28
>
> We thank the reviewer for their insightful comments. We address the key points below; please let us know if there is anything else that we can clarify. All references are included in our response to reviewer WxGU.
>
> **W1**
>
> Assumption 1 (which we now term “Public Information”) is restrictive but standard in settings with public information (e.g. stock prices in finance, interest rates and wages in macroeconomics, infection statistics in epidemiology, and signal in communications [1-5]), our applications of interest. In macroeconomics, for example, agents observing common prices is a standard “competitive equilibrium” assumption. We have clarified our applications and assumptions in the introduction.
>
> **W2**
>
> The reviewer’s insight is exactly correct: our contribution is “about showing that when white-box access is available, one should exploit it.” Fully sample-based RL methods are model-free [6] and treat transition dynamics as a black box, failing to leverage the fact that, under modeling assumptions, these dynamics are accessible. As mentioned to reviewer A6kJ, we have revised our introduction to distinguish our problem setting from standard RL. Using white-box access for best-response computation does not introduce asymmetry: a black-box RL method conditioning on the same information could, in principle, match its performance given sufficient samples.
>
> **W3**
>
> While current prices appear to provide most of the information needed for optimal control in the Krusell-Smith environment (explaining similar exploitability for SPG and RSPG), this was not obvious a priori and is not implied by prices being deterministic functions of the mean field, since prices alone are not Markovian. In fact, one motivation for our work was precisely to evaluate whether the assumption of memoryless policies in the SPG paper [14] was limiting. Despite this, the environment still validates that history-aware behaviour is learned with RSPG: time is unobserved, but Figure 3 still shows that agents learn anticipatory behaviour (spending more closer to the end of the episode), which is not the case for SPG. More generally, all three environments validate that RSPG captures history-dependence without compromising proximity to equilibrium (in fact, RSPG often achieves lower exploitability).
>
> **KQ1**
>
> We have not empirically evaluated RSPG under violations of Assumption 1. As discussed in Section 6.2, in such cases we recommend using fully sample-based methods that allow policies to condition on individual histories, as HSMs become intractable. Nonetheless, were one to train the policy under state-independent observations using RSPG but deploy in an environment with state-dependent ones, we would generally expect exploitability to increase due to training the policy in a misspecified MFG. We expect degradation to be problem-dependent rather than uniformly catastrophic, with small violations (e.g. observations being noisier than those assumed during training) leading to gradual performance loss. That being said, many applications assume shared public information [1-5], where RSPG remains applicable and offers significant computational advantages. We also remark that, even in cases with individual observations, public belief models have been introduced to efficiently improve policies in partially observable settings [25].
>
> **KQ2**
>
> The reviewer's insight is exactly correct: under Assumption 1, the GRU hidden state is identical for all agents at a given timestep. In our implementation we do exactly as suggested, and amortise this by computing a single recurrent aggregate embedding per timestep and broadcasting it to all agents. As a result, the additional cost of RSPG over SPG is just one GRU update per timestep, independent of the number of states (note that in HSMs, rather than tracking agents, we track states). Both methods therefore scale identically with the number of states, with RSPG incurring only a constant-factor overhead relative to the MLP embedding used in SPG.
>
> **KQ3**
>
> We agree that a discussion of convergence is important. As mentioned, existing guarantees for MFG algorithms rely on strong assumptions, such as full-observability, monotonicity, and continuous reward and transition functions [17, 12, 18, 26], which do not hold in our applications. [23] establish convergence for independent learning in partially observable MFGs with shared aggregate observations, but only for memoryless policies. Extending such guarantees remains a direction for future research: we have added a discussion explicitly acknowledging this in our work. In principle, RSPG could be combined with regularisation methods (e.g. Fictitious Play [1]), although we did not find this necessary in our experiments; empirically, we observe stable convergence throughout (e.g., Figure 2). We hypothesise that the variance reduction introduced by HSMs improves the stability of gradient estimates and contributes to the observed convergence behaviour.

---

> > ### Author Rebuttal · Reviewer_eZCK · 2026-04-02
> >
> > Thank you for addressing all the questions and weaknesses I have raised in the review in a precise way. I think that the adjustment will strengthen the paper's contribution.
> > I will adjust my score accordingly.

---

### Official Review · Reviewer_A6kJ · 2026-03-18

**Soundness:** 4
**Presentation:** 3
**Significance:** 4
**Originality:** 3
**Overall Recommendation:** 5
**Confidence:** 5

**Summary:**

This work is concerned with how to solve partially observable mean-field games with common noise while retaining the variance-reduction benefits of structural methods. The paper introduces Recurrent Structural Policy Gradient (RSPG), a “history-aware” hybrid structural method that replaces full individual-action-observation histories with shared public observation histories under a compressed-observability assumption, and pairs this with a new JAX framework, MFAX. The paper reports order-of-magnitude wall-clock gains over RL baselines and strong exploitability results on Linear Quadratic, Beach Bar, and a macroeconomics environment.

**Compliance With Llm Reviewing Policy:**

Affirmed.

**Key Questions For Authors:**

The paper would benefit from a short roadmap, possibly in the introduction or appendix, that separates three sources of difficulty: mean-field interaction, common noise, and partial observability. Right now these are introduced together, which makes it hard for non-experts to disentangle what each ingredient changes mathematically and algorithmically. The survey literature would be a natural place to anchor such a discussion.

Please elaborate more clearly on why the compressed-observability / shared-public-history setting is the right application model.

Finally, the literature review around algorithms for MFGs could be broadened substantially. Work by Mathieu Laurière, René Carmona, Romuald Élie, and co-authors has already developed a fairly extensive algorithmic literature for mean-field control and games and a more detail overivew, even if in the appendix of key ideas in the field would be useful when arguing novelty.

**Limitations:**

yes

**Strengths And Weaknesses:**

Strengths:
The paper tackles an ambitious and important setting: mean-field games with both partial observability and common noise. This is a genuinely difficult combination. The  common noise does not average out at the population level, so the aggregate state remains stochastic. The technical device used in Section 6 — reducing tractability issues by restricting to shared observations and proving equivalence to a reduced policy class under Assumption 1 — is elegant and, within that setting, useful. The empirical section is also reasonably convincing that the proposed structural method can outperform purely sample-based RL baselines in wall-clock time and exploitability on the chosen benchmarks.

Weaknesses:
My main concern is positioning and novelty clarity. The paper addresses a broad and already active area, but the related-work discussion is not yet rich enough for a non-expert reader to understand what is established in the literature, what is difficult specifically because of common noise, what is difficult specifically because of partial observability, and what is genuinely new here. There is already extensive work on learning in MFGs, including a broad survey by Laurière et al., neural and deep-learning methods for mean-field control and games by Carmona and Laurière, and a broader review of machine-learning methods for stochastic control and games by Hu and Laurière. The current manuscript cites some nearby papers, but I do not think it gives the reader an adequate map of this landscape.

Relatedly, the partial-observability aspect needs to be positioned more precisely. In classical partially observed control, without additional structure, optimal policies typically depend on the information state or belief induced by the entire observation history. Here, the key tractability result comes from Assumption 1, namely that the observation of the aggregate state is independent of the individual state, which the paper explicitly connects to “Compressed Observability.” That assumption may well be sensible in some applications, but it is also a strong restriction.

A second issue is the “hybrid structural” assumption itself. The method requires known individual transition dynamics and tractable push-forward updates, and the paper acknowledges this trade-off explicitly. That is a reasonable modeling choice in some macroeconomic or finance-style applications, and the macroeconomics experiment is interesting in that regard. But in many applications transition kernels are not known and must be estimated or learned. As such this is not a surprise that the methods perform better than pure RL

---

> ### Author Rebuttal · Authors · 2026-03-28
>
> Firstly, we thank the reviewer for their encouraging and constructive feedback. We have revised the introduction and related work to explicitly state our applications and assumptions, and clearly position our contribution within the broader MFG literature. We summarise the key additions below; please let us know if there is anything else that we can clarify. All references are included in our response to reviewer WxGU.
>
> **Applications and Assumptions.**
>
> To address the reviewer’s questions regarding the assumptions, although they may be restrictive, they are standard in many settings with public information, which are our applications of interest. Clearly stated, in settings such as macroeconomics, financial markets, epidemic control, communication systems and power networks [1-5], the following two structural properties are natural. First, the individual state space is typically low dimensional (such as wealth and income in macroeconomics and finance, infection status in epidemiology, or energy and interference in communication systems), while the common noise and hence aggregate state can be arbitrarily high dimensional. Second, as well as knowing their individual state, agents receive shared observations of the aggregate state, corresponding to public information (such as prices, interest rates or infection statistics). Finally, we distinguish these applications (with access to underlying model dynamics and where the aim is to compute equilibrium behaviour in a known system), from standard RL, where dynamics are unknown and must be learned from data.
>
> **Sources of Difficulty.**
>
> To clarify the positioning of our work, we distinguish three orthogonal challenges in MFGs: first, mean-field interaction, where agents interact through the population distribution; second, common noise, which introduces stochastic evolution of the mean-field; and third, partial-observability, where agents must act based on incomplete information. Existing methods typically address only subsets of these challenges. Our work focuses on the joint setting, where all three are present in the aforementioned applications.
>
> **Extended Related Works.**
>
> Considerable work on MFGs already exists. [6] survey the standard setting (excluding common noise and partial observability), where the mean-field evolves deterministically. Numerical methods [7, 8] can solve these problems when analytic solutions are unavailable, but are not suitable for settings with common noise. Solving the resulting Master Equation [9], has motivated the use of Deep Learning-based methods [10, 11].
>
> While deep RL methods have been developed for MFGs with common noise, these methods are model-free [6], treating dynamics as a black box and foregoing variance reduction from known structure. Although model-based methods have been proposed [12], these require enumerating realisations of common noise and associated mean-field trajectories, limiting scalability. Recent Hybrid Structural Methods [13], such as SPG [14], leverage known structure to achieve lower-variance updates by integrating over individual dynamics and maintain tractability by sampling common noise.
>
> Furthermore, most RL-based algorithms for MFGs assume full observability, with policies conditioning on the local state (without common noise) [15, 16, 17, 18] or both the local and aggregate state (with common noise) [19, 20]. Partial observability has been considered in limited forms: [21] restrict observations to local neighbourhoods, and [22] allow agents to estimate the mean-field, but both use memoryless policies. On the theoretical side, [23] study independent learning in partially observable MFGs with shared observations and prove convergence for memoryless policies, while [24] consider more general observation kernels. Neither includes common noise. Because (to our knowledge) this is the first work to address both common noise and partial observability, we formalise the problem setting using a general observation model, and only then consider the special case of shared aggregate observations.
>
> Our key contribution is extending HSMs to support history-aware policies while preserving tractability. Existing HSMs are limited to tabular, memoryless policies; in our applications, agents receive shared observations of the aggregate state, which means that conditioning only on current observations discards useful information from the past. The main insight is that, since individual states are known and aggregate observations are shared, memory can be restricted to the history of shared observations; intuitively, agents must only maintain a belief in the aggregate state. As a result, RSPG benefits from variance reduction compared to fully model-free methods, while leveraging neural network function approximation to handle memory, and a shared history to maintain tractability.

---

> > ### Author Rebuttal · Reviewer_A6kJ · 2026-03-31
> >
> > Thank you for providing the answer. I acknowledge that the setting og the paper is very general, but I still think novelty, apart from mixing together results from literature, needs to be better articulated.

---

> > > ### Author Response · Authors · 2026-04-01
> > >
> > > Thank you for your prompt response and helpful feedback. We have summarised our key contributions into three main points (below) and revised the introduction and related work section to more clearly articulate the novelty of our work and distinguish it from existing literature. We hope that this fully addresses your final concern.
> > >
> > > **Contribution 1: Methodological**
> > >
> > > In its most succinct form, our main contribution is a tractable method for computing analytic mean-field updates using history-conditioned policies. This has several implications, the most important being that it enables differentiation through the individual state transitions, allowing us to retain the variance reduction that makes Hybrid Structural Methods significantly faster than fully sample-based RL methods, while still learning history-conditioned policies that produce more realistic behaviour in partially-observable mean-field games. Note, also, that this work is the first to scale HSMs to policies parameterised by neural network function-approximators, which is essential for partially observable environments and a non-trivial number of realisations of common-noise.
> > >
> > > **Contribution 2: Formalism of Partially Observable Mean Field Games with Common Noise**
> > >
> > > To the best of our knowledge, this is the first work to formally define partially observable mean-field games with common noise. This is why we define them in the more general setting (using policies conditioned on the full history), and only then consider the special case of shared aggregate observations (relevant to our applications of interest). We are not aware of prior work (whether sample-based RL methods or HSMs) that learns history-conditioned policies for MFGs. No prior work explicitly identifies the difficulty of updating the mean-field in this setting (whether using a sample-based approach, or using an analytic update).
> > >
> > > **Contribution 3: Open Source MFG Library**
> > >
> > > We also introduce and open-source MFAX, our library for MFGs that makes three main improvements over current MFG libraries. First, by employing a functional representation of the mean-field update that exploits sparsity, MFAX is substantially faster than current libraries. Second, unlike existing libraries (which are limited to fully observable environments without common noise) MFAX supports partially observable environments with common noise. Third, as noted in response to reviewer WxGU, MFAX clearly separates different MFG problem settings (standard RL problem settings (treating transition dynamics as unknown) from modelling problem settings (that can leverage known dynamics)) by providing wrappers for both exact mean-field updates (required for HSMs when differentiating through expected returns over next-state transitions), and sample-based mean-field updates (necessary when exact updates are intractable in high-dimensional state spaces, for example). We hope that MFAX will facilitate further research into more complex MFGs.
> > >
> > > In summary and to the best of our knowledge, this is the first work to explicitly tackle MFGs that are both partially observable and with common noise.

---

### Decision · Program_Chairs · 2026-04-30

**Decision:**

Accept (spotlight)

**Comment:**

The reviewers are all positive and the rebuttal have resolved main concerns. The paper makes a meaningful contribution by extending hybrid structural methods to history-aware policies in partially observable mean-field games with common noise, and the empirical results are strong. The main limitations (especially the restrictive public-information assumption and the stronger modeling assumptions relative to model-free RL baselines) are real, but they do not undermine the paper’s technical contribution. Overall, I recommend acceptance.